# Are the Common Genetic 3′UTR Variants in ADME Genes Playing a Role in Tolerance of Breast Cancer Chemotherapy?

**DOI:** 10.3390/ijms252212283

**Published:** 2024-11-15

**Authors:** Karolina Tęcza, Magdalena Kalinowska-Herok, Dagmara Rusinek, Artur Zajkowicz, Aleksandra Pfeifer, Małgorzata Oczko-Wojciechowska, Jolanta Pamuła-Piłat

**Affiliations:** Department of Clinical and Molecular Genetics, Maria Sklodowska-Curie National Research Institute of Oncology, Gliwice Branch, 44-102 Gliwice, Poland; karolina.tecza@gliwice.nio.gov.pl (K.T.); magdalena.kalinowska-herok@gliwice.nio.gov.pl (M.K.-H.); dagmara.rusinek@gliwice.nio.gov.pl (D.R.); artur.zajkowicz@gliwice.nio.gov.pl (A.Z.); aleksandra.pfeifer@gliwice.nio.gov.pl (A.P.); malgorzata.oczko-wojciechowska@gliwice.nio.gov.pl (M.O.-W.)

**Keywords:** breast cancer, SNPs, toxicity, chemotherapy

## Abstract

We studied the associations between 3′UTR genetic variants in ADME genes, clinical factors, and the risk of breast cancer chemotherapy toxicity. Those variants and factors were tested in relation to seven symptoms belonging to myelotoxicity (anemia, leukopenia, neutropenia), gastrointestinal side effects (vomiting, nausea), nephrotoxicity, and hepatotoxicity, occurring in overall, early, or recurrent settings. The cumulative risk of overall symptoms of anemia was connected with *AKR1C3* rs3209896 AG, *ERCC1* rs3212986 GT, and >6 cycles of chemotherapy; leukopenia was determined by *ABCC1* rs129081 allele G and *DPYD* rs291593 allele T; neutropenia risk was correlated with accumulation of genetic variants of *DPYD* rs291583 allele G, *ABCB1* rs17064 AT, and positive HER2 status. Risk of nephrotoxicity was determined by homozygote *DPYD* rs291593, homozygote *AKR1C3* rs3209896, postmenopausal age, and negative ER status. Increased risk of hepatotoxicity was connected with *NR1/2* rs3732359 allele G, postmenopausal age, and with present metastases. The risk of nausea and vomiting was linked to several genetic factors and premenopausal age. We concluded that chemotherapy tolerance emerges from the simultaneous interaction of many genetic and clinical factors.

## 1. Introduction

Chemotherapeutic medicines commonly used in breast cancer therapy are fraught with high risk of side effects and toxicity during and after therapy. Chemotherapy affects highly proliferative cells, both cancerous and normal. Moreover, cytotoxic treatment also affects the organs and tissues with low proliferatory capabilities, but crucial for the organism’ detoxication [1,2]. Also, because of the systemic impact of anticancer drugs, the multisystem organ failure is one of the possible adverse effects, and, consequently, remains a strong natural therapy-limiting factor [3,4,5].

Myelotoxicity is a frequent side effect observed during chemotherapy that leads to anemia, neutropenia, leukopenia, and thrombopenia [6]. Among the normal cells, bone marrow progenitors are often targeted, which results in a decrease in blood cells production [1,2]. Treatment delays and dose reduction in patients with neutropenic or leukopenic events may also further compromise the prognosis for these patients [2,7,8]. Chemotherapy-induced nausea and vomiting (CINV) is more often noted in younger patients [9,10]. These symptoms may lead to disruptions in cancer treatments and to severely decrease the patient’s quality of life. Still, the pathophysiology of CINV and gastrointestinal side effects is not fully known [11,12]. Nephrotoxicity results from a rapid deterioration in kidney function due to the toxic effects of drugs. These events are the natural consequence of the kidneys’ role in homeostasis, excretion of toxic metabolites, detoxification, and regulation of extracellular fluids [13]. Kidney toxicity is frequent in the elderly due to an increase in the life span and polymedications [14]. Hepatotoxicity induced by overload of drugs is a rare symptom observed during treatment. However, suspension of therapy is most often the result of an increase in liver enzymes. Drug-induced hepatotoxicity can result from direct toxicity of drugs, but also indirectly from a reactive metabolite or from immunologically-mediated response affecting hepatocytes, biliary epithelial cells, and/or liver vasculature [15,16,17]. Hepatotoxicity’s manifestations are diversified, and range from asymptomatic elevation of liver enzymes, weakness, nausea or vomiting, severe fatigue, and severe abdominal pain to fulminant hepatic failure [18,19].

Significant variability in drug response may occur among breast cancer patients treated with the same medications with various intensity. Recently, pharmacogenomic studies have elucidated the inherited nature of these differences in drug effects and showed that functional single nucleotide polymorphisms (SNPs) are one of the possible causes of the interindividual differences in clinical outcomes among breast cancer patients [20,21,22]. Moreover, SNPs in genes encode enzymes of the absorption, distribution, metabolism, and excretion correlated to treatment toxicity [23,24,25]. Anticancer drugs (doxorubicin, cyclophosphamide) commonly used in breast cancer patients are metabolized by hydroxylation enzymes (carbonyl reductase CBRs genes), semiquinone formation (*NDUFS*, *NQO1*, *XDH*, *NOS* genes), deoxyaglycone formation (*POR*, *XDH*, *NQO1* genes), metabolisms enzymes phase I and phase II (cytochrome P450, ADH, and ALDH), glutathione S-transferase (GST), sulfotransferase (SULT), and membrane transporters (influx transporters SLC and efflux ABC transporters family) [22]. These proteins play important roles in conferring multidrug resistance in cancer cells, as well as in alternation of clearance of chemotherapy drugs with enhanced toxicity. The interpatient difference in the efficiency of the detoxification and transport system can modulate the therapeutic effect. A small increase in plasma drug concentration may lead to toxicity, while a small decrease in concentration may reduce treatment effectiveness [26]. Polymorphisms (SNPs) in noncoding regulatory regions of the gene can affect gene splicing and transcription factor binding [27,28,29,30]. SNPs in regulatory 3′UTR sequences may create or abolish an miRNA target and disrupt mRNA expression. Polymorphisms in 3′UTRs of ADME genes lead to different levels of enzyme activity. SNPs in influx and efflux transporter genes influence correct drug intake [24].

In our previous study, we described a correlation between SNPs in 3′UTR of ADME genes and survival in breast cancer patients [31]. In the current study, we report a correlation of germline SNPs in 3′UTRs of ADME genes and the risk of toxicity and side effects of breast cancer chemotherapy. The analyzed genes are involved in the metabolism and transport as well as the activity of the cellular repair system in the polychemotherapy regimen (FAC) based on anthracycline (doxorubicin), alkylate drug (cyclophosphamide), and antimetabolite (5′-fluorouracil). The polymorphisms in breast cancer patients were tested in relation to seven symptoms belonging to myelotoxicity, gastrointestinal side effects, nephrotoxicity, and hepatotoxicity.

## 2. Results

### 2.1. Association of SNPs and Clinical Factors with Anemia

Risk of overall anemia inducted by FAC chemotherapy was correlated with the presence of genetic and clinical factors: *AKR1C3* rs3209896 heterozygous variant AG (OR 3.00; 1.23–7.26; *p* = 0.015), *ERCC1* rs3212986 genotype GT (OR 2.50; 1.09–5.75; *p* = 0.030), and number of chemotherapy cycles above six (OR 3.88; 1.48–10.16; *p* = 0.005) (Table 1). In the cumulative analyses, the presence of a growing number of high-risk factors was reflected in the increasing risk of overall anemia from OR 3.70; 1.06–12.87; *p* = 0.038 for the two factors, to OR 30.83; 5.22–181.97; *p* = 0.0001 for the carriers of all three of them. The carriers of all three factors had a more than twenty times higher risk of overall anemia (OR 20.30; 4.52–91.18; *p* = 0.00008) than the combined 0’s–2’s reference groups (The second table in Section 2.7).

The anemia symptoms that appeared in the first two cycles of FAC chemotherapy (early anemia) were associated with the presence of three genetic factors: *ABCC1* rs129081 common homozygote GG (OR 3.52; 1.018–10.51; *p* = 0.023), *AKR1C3* rs3209896 variant AG (OR 5.85; 1.56–21.83; *p* = 0.008), and *RALBP1* rs12680 allele C (OR 4.41; 1.42–13.74; *p* = 0.009) (Table 1). When taken together, the lack of any high-risk factors (i.e., the noncarrier group) was reflected in the absence of this symptom in the first two cycles of FAC chemotherapy (The second table in Section 2.7). Due to this fact, the calculations of the early anemia cumulative risk used the group combined from noncarriers and carriers of only one risk factor as the reference. The presence of all three unfavorable factors increased the risk of early anemia by more than seventeen fold (OR 17.28; 2.65–12.58, *p* = 0.003). Risk for 2’–3’ factors carriers increased by nearly eight times vs. reference group (OR 7.79; 2.59–23.38; *p* = 0.0002). Similar results were observed for carriers of all three high-risk factors in comparison to 0’–2’ reference groups (OR 7.77; 1.37–43.95; *p* = 0.02).

The risk of anemia present at four or more cycles of FAC treatment (recurrent anemia) was determined by the occurrence of *ABCC1* rs129081 genotype GG (OR 8.02; 0.02–0.63; *p* = 0.012), *ERCC1* rs1046282 heterozygote TC (OR 5.24; 1.0–6.99; *p* = 0.047), and *UGT2B4* rs1131878 variant AA (OR 8.91; 0.01–0.94; *p* = 0.043) (Table 1). In the cumulative model, we observed higher symptom risk correlated to an increased number of adverse factors. Also, the noncarrier group did not develop the studied symptoms. Carriers of three high-risk genotypes were over fifty-four times more likely to have recurrent anemia when compared to the reference group (OR 54.41; 5.03–499.36; *p* = 0.0004). The risk for carriers of two and/or three high-risk factors (2’ and/or 3’) was OR 14.63; 1.79–119.85; *p* = 0.012 vs. reference 0’–1’. The risk of symptom for the carriers of three unfavorable factors was OR 19.78; 4.83–80.91; *p* = 0.00003, in comparison to the 0’–2’s reference group (The second table in Section 2.7).

### 2.2. Association of SNPs with Leukopenia

In multivariate analysis risk, symptoms of overall leukopenia were conditioned by two genetic factors: *ABCC1* gene rs129081 allele G (OR 1.89; 0.27–0.99; *p* = 0.048) and *DPYD* rs291593 allele T (OR 1.73; 1.08–2.76; *p* = 0.020) (Table 2). In the cumulative analysis, we observed statistically significantly high symptom risk for carriers of both genetic factors (OR 3.84; 1.37–10.76; *p* = 0.01) when compared to noncarriers, and OR 1.81; 1.13–2.89; *p* = 0.013 in relation to other remaining patients. Similarly, the group of patients with 1’–2’ adverse factors showed a higher risk of FAC-related leukopenia in comparison to noncarriers (OR 2.92; 1.09–7.86; *p* = 0.033) (The second table in Section 2.7).

The occurrence of leukopenia symptoms in the two first FAC chemotherapy cycles (early leukopenia) was determined by *DPYD* rs291583 variant GG (OR 2.25; 1.25–4.05; *p* = 0.006) and *AKR1C3* rs3209896 genotypes AA and GG (OR 1.77; 1.08–2.89; *p* = 0.021) (Table 2). Cumulative analyses showed an increased risk of symptoms accompanied by an increase in the number of adverse variants. The presence of one independent risk factor for early leukopenia was responsible for nearly twofold elevation of this symptom’s risk (OR 1.97; 1.16–3.37; *p* = 0.012), and the presence of two independent risk factors was responsible for a more than three times increased risk (OR 3.75; 1.55–9.03; *p* = 0.003). Similar to previous analyses, a group of patients with 1’ and 2’ unfavorable genetics factors showed a higher risk of FAC-related early leukopenia in comparison to noncarriers (OR 2.17; 1.29–3.65; *p* = 0.003). In the cumulative analysis the 2’s vs. patients without or with just one factor (reference group 0’–1’) had a higher chance of early leukopenia (OR 2.45; 1.10–5.49; *p* = 0.027) (The second table in Section 2.7).

The risk of recurrence of leukopenia in four or more cycles was increased by two independent genetic factors: *ABCA1* rs4149339 common and rare homozygotes CC/TT (OR 2.66; 0.14–0.95; *p* = 0.04) and *DPYD* rs291583 homozygote GG (OR 2.80; 1.26–6.19; *p* = 0.011) (Table 2). In the cumulative model, the carriers of both those factors had a more seven times higher risk of this symptom than the noncarriers (OR 7.14; 2.05–24.91; *p* = 0.002). Carriers 1’–2’ of adverse factor had a more than three times higher risk of recurrent leukopenia than the reference group (OR 3.13; 1.06–9.26; *p* = 0.039). This trend was confirmed by comparison of carriers of 2’ unfavorable factors vs. reference 0’–1’ carriers (OR 3.75; 1.61–8.75; *p* = 0.002) (The second table in Section 2.7).

### 2.3. Association of SNPs and Clinical Factor with Neutropenia

The *ABCB1* rs17064 heterozygous variant (AT) (OR 4.56; 1.25–16.63; *p* = 0.021), *DPYD* rs291583 allele G (OR 1.83; 1.03–3.21; *p* = 0.036) and positive HER2 status (OR 1.69; 0.99–2.87; *p* = 0.049) increased overall neutropenia risk (Table 3). The simultaneous presence of three adverse factors further increased the symptom’s risk (OR 14.00; 1.38–142.44; *p* = 0.020) (The second table in Section 2.7). Genetic variants established as independent risk factors for early neutropenia were *GSTM3* rs3814309 genotype CC (OR 3.12; 1.29–7.52; *p* = 0.01), *ABCB1* rs17064 variant AT (OR 3.52; 1.29–9.56; *p* = 0.013), *ERCC1* rs1046282 genotype CC (OR 5.88; 1.50–22.97; *p* = 0.01), and *ALDH5A1* rs1054899 allele C (OR 1.66; 1.0–2.75; *p* = 0.046) (Table 3). The presence of adverse factors increased early neutropenia risk (OR 2.38; 1.39–4.10; *p* = 0.002 and OR 4.88; 2.01–11.81; *p* = 0.0004), respectively, compared to the reference group. In the studied group, there was only one patient with three unfavorable variants; therefore, the risk calculation in the regression model was not possible. In the cumulative model, the risk of early neutropenia increased by over fivefold (OR 5.18, 2.16–12.55, *p* = 0.0002) in carriers of 2’–3’ adverse factors compared to noncarriers. The risk of early neutropenia in carriers of any number of adverse factors was also elevated (OR 2.71; 1.61–4.57; *p* = 0.0002). In addition, carriers of 2’–3’ of adverse factors had a more than three times increased risk of the symptom compared to the reference group (OR 3.14, 1.40–7.02, *p* = 0.005) (The second table in Section 2.7). The risk of neutropenia in four or more cycles of FAC chemotherapy (recurrent neutropenia) was determined by the presence of the *ABCC1* rs212091 allele AG (OR 3.14; 1.36–7.25; *p* = 0.007), *UGT2B4* rs1131878 variant AG (OR 2.68; 1.13–6.34; *p* = 0.024), and positive status of PR (OR 2.65; 1.04–6.68; *p* = 0.039) (Table 3). In the cumulative analysis, the higher incidence of recurrent neutropenia was observed in the carriers of three high-risk genotypes (OR 17.69; 1.81–172.51; *p* = 0.012). The risk of the studied symptoms in the combined calculation of 3’ vs. 0’–2’ carriers was established at OR 4.28; 1.38–13.08; *p* = 0.011 (The second table in Section 2.7).

In our study, the risk of severe recurrent neutropenia (grades 3 and 4) was independently influenced by the *ABCB1* rs17064 genotype AT (OR 5.13; 1.38–19.02; *p* = 0.014), *UGT2B4* rs1131878 genotype AG (OR 3.78; 1.26–11.24; *p* = 0.016), and *ALDH5A1* genotype AC (OR 3.94; 1.34–11.53; *p* = 0.012) (Table 3). The simultaneous occurrence of all three above-listed risk factors increased the risk of severe recurrent neutropenia over ninety-fold when compared to the noncarriers group (OR 91.0; 5.45–1520.36; *p* = 0.002). This effect was confirmed in comparison of the 3’s group vs. the group combined from 0’s–2’s (OR 16.35; 2.15–124.32; *p* = 0.007) (The second table in Section 2.7).

### 2.4. Association of SNPs and Clinical Factors with Nausea

The presence of NOS3 rs2566508 homozygotes TT and GG (OR 1.73; 1.00–2.91; *p* = 0.038), CYP1B1 rs162562 genotype AA (OR 1.63; 1.00–2.66; *p* = 0.048), DPYD rs291583 allele G (OR 1.90; 1.12–3.21; *p* = 0.015), and premenopausal age (OR 2.98; 1.12–3.21; *p* = 0.040) increased the risk of early nausea (Table 4). In the cumulative model, we observed that carriers of more than two risk factors had an over two times higher risk of early nausea in comparison to the carriers of two or fewer factors (reference group) (OR 1.84; 1.12–3.00; *p* = 0.014) (The second table in Section 2.7).

The risk of early severe nausea was determined by the *AKR1C3* rs3209896 variant AG and patient age under 39 years (OR 3.80; 1.26–11.39; *p* = 0.016 and OR 7.49; 2.23–25.08; *p* = 0.001, respectively) (Table 4). In the cumulative analyses, for carriers of both factors, the calculated risk reached OR 34.5; 6.18–192.60; *p* < 0.00001 when compared to noncarriers, and OR 18.5; 4.19–81.91; *p* = 0.0001 in relation to the noncarriers and the presence of single independent risk factor combined. The presence of any number of determinants of early severe nausea vs. noncarriers was responsible for toxicity risk at the level of OR 3.59; 1.15–11.15; *p* = 0.026 (The second table in Section 2.7).

The risk of recurrent nausea was correlated with the presence of multiple genetic independent factors: *ERCC4* rs2276464 allele C (OR 2.63; 1.26–5.48; *p* = 0.010), *SULT4A1* rs138057 variant GG (OR 3.76; 1.16–12.17; *p* = 0.027), *DPYD* rs291593 variant CT (OR 2.65; 1.27–5.59; *p* = 0.010), *NOS3* rs2566508 homozygotes TT and GG (OR 2.74; 1.07–7.06; *p* = 0.036), and *ALDH5A1* rs1054899 homozygote AA (OR 4.72; 1.67–13.36; *p* = 0.003) (Table 4). In the cumulative analyses, we observed a statistically significant increase in symptom risk for the carriers of three factors (OR 8.12; 3.12–20.66; *p* = 0.00001) in comparison to the reference group constructed from the 0’s and 1’s. Similarly, a higher risk of recurrent nausea was detected in carriers 2’s–4’s of unfavorable factors (OR 3.74; 1.59–8.80; *p* = 0.0024) vs. the carriers of 0’–1’ factors (reference group) (The second table in Section 2.7). It should be noted that in the studied group, recurrent nausea was absent in the noncarrier group; also, we did not detect any carrier of all five high-risk factors.

The severe nausea was associated with the presence of *UGT2B15* rs3100 common homozygote CC (OR 2.31; 1.11–4.81; *p* = 0.025) and premenopausal age (OR 3.52; 1.26–10.12; *p* = 0.019) (Table 4). The cumulative analysis showed an increased risk of severe nausea with the accumulation of both those factors, from OR 2.33; 1.08–5.03; *p* = 0.030 for one factor to OR 8.51; 2.11–34.33; *p* = 0.009 for the presence of all two. In the cumulative model, a nearly threefold increase in this symptom’s risk was detected for groups 1’–2’ combined (OR 2.71;1.29–5.67; *p* = 0.008) in relation to 0’–1’carriers as the reference. The presence of two risk factors increased nausea severe risk by more than fivefold (OR 5.76; 1.53–21.67; *p* = 0.009) (The second table in Section 2.7).

### 2.5. Association of SNPs and Age with Vomiting

The risk of overall vomiting was independently correlated with the presence of *ABCC5* rs3805114 common homozygote AA (OR 2.64; 1.05–6.62; *p* = 0.037) and premenopausal age (OR 3.60; 1.38–9.42; *p* = 0.009) (Table 5). In the cumulative model, the carriers of two high-risk factors had a more than ninefold increased risk of overall vomiting compared to the reference noncarriers group (OR 9.51; 2.37–38.17; *p* = 0.0012). The presence of one or two independent risk factors for overall vomiting was responsible for a nearly threefold elevation of this symptom’s risk (OR 2.92; 1.10–7.78; *p* = 0.03 and OR 4.01; 1.43–11.22; *p* = 0.008) (The second table in Section 2.7).

The *RALPB1* rs12680 allele C (OR 2.05; 0.97–4.32; *p* = 0.057; result at borderline significance) and premenopausal age (OR 2.92; 1.11–7.65; *p* = 0.028) were correlated with a higher risk of early vomiting (Table 5). The carriers of two those risk factors had an increased toxicity risk to OR 20.3; 2.18–188.49; *p* = 0.008 when compared to the noncarriers, and also to the 0’s and 1’s combined (OR 18.26; 1.98–168.25; *p* = 0.010) (The second table in Section 2.7). Also, the very presence of any risk determinants resulted in a higher chance of early vomiting (OR 2.15; 1.11–2.15; *p* = 0.023).

The allele T of *ABCB1* rs17064 (OR 8.63; 1.37–54.56; *p* = 0.021) and genotype GG of *NR1/2* rs3732359 (OR 6.44; 1.19–34.77; *p* = 0.030) were correlated with recurrent vomiting (Table 5). Cumulative analysis showed extremely high risk for the carriers of two unfavorable genetic factors for comparison to the noncarriers (OR 59.00; 3.63–959.44; *p* = 0.004), as well as to the other groups combined (OR 29.00; 2.22–378.35; *p* = 0.010) (The second table in Section 2.7). However, it should be noted that in our group of patients we detected only one carrier of two unfavorable genetic factors with recurrent vomiting symptoms. The presence of any combination of high-risk factors resulted in an increase in risk by more than eight times (OR 8.43; 1.50–47.50; *p* = 0.015).

Risk of severe vomiting was correlated with the presence of allele T of *ABCB1* rs17064 (OR 6.46; 1.65–25.25; *p* = 0.007), *SULT4A1* rs138057 allele AA (OR 6.50; 1.37–30.76; *p* = 0.017) and premenopausal age (OR 8.36; 2.07–33.75; *p* = 0.003) (Table 5). In the cumulative model, the carriers of two high-risk factors are nearly twenty-five times more likely to develop severe vomiting (OR 24.46; 4.39–136.2644; *p* = 0.0002) in comparison to the reference group of noncarriers. Also, the presence of two or more risk factors resulted in a highly significant elevation of risk for severe vomiting (OR 15.97; 5.12–49.87; *p* < 0.00001) (The second table in Section 2.7). The risk of severe early vomiting was associated with *SULT4A1* rs138057 allele G (OR 11.34; 1.31–98.06; *p* = 0.026) and premenopausal age (OR 14.50; 3.38–62.19; *p* = 0.0003) (Table 5). In the cumulative analysis, the risk of toxicity for carriers of two unfavorable factors was extremely high (OR 92.00; 8.42–1004.90; *p* = 0.0002) in opposition to noncarriers and also in reference to the 0’–1’ group (OR 32.11; 7.02–146.81; *p* < 0.00001) (The second table in Section 2.7).

### 2.6. Association of SNPs and Clinical Factors with Nephrotoxicity

The risk of overall nephrotoxicity was correlated with the presence of *DPYD* rs291593 genotypes CC and TT (OR 7.23; 1.44–36.14; *p* = 0.016), *AKR1C3* rs3209896 genotypes AA and GG (OR 6.71; 1.33–33.75; *p* = 0.02), postmenopausal age (OR 7.57; 2.12–26.98; *p* = 0.002), and negative status of ER (OR 5.71; 1.14–18.89; *p* = 0.012) (Table 6). In the cumulative model, the carriers of no and one high-risk factors did not develop nephrotoxicity, so the reference group was constructed from three carrier subgroups: 0’s–2’s. In this setting, the carriers of all four high-risk factors had an over fifty-three-fold increased risk of overall nephrotoxicity (OR 53.25; 5.85–484.25; *p* = 0.0004). The presence of three and four or all four independent risk factors for overall nephrotoxicity was responsible for a nearly twenty-nine-fold elevation of this symptom’s risk (OR 28.40; 6.10–132.21; *p* = 0.00001 and OR 10.58; 1.75–64.12; *p* = 0.01) (The second table in Section 2.7).

The risk of early nephrotoxicity in multivariate analysis was correlated with the presence of genetic factors: *ABCC5* rs3805114 variant AC (OR 6.07; 1.11–32.99; *p* = 0.035), *ERCC4* rs4781563 genotype AA (OR 24.66; 2.22–273.1; *p* = 0.008), *DPYD* rs291583 genotype CC (OR 14.92; 1.13–195.99; *p* = 0.038), and perimenopausal age (OR 10.25; 1.11–94.14; *p* = 0.038) (Table 6). In the cumulative analysis, carriers of three unfavorable factors had a nearly five-fold increased risk of nephrotoxicity in the two first cycles of chemotherapy (OR 4.88; 2.30–10.38; *p* = 0.00003) vs. the combined 0’–1’ factors reference group. In subsequent cumulative calculation, the carriers of three unfavorable factors had a nearly forty-seven-fold increased risk of nephrotoxicity in two first cycles of FAC chemotherapy (OR 46.66; 9.55–227.80; *p* < 0.00001). In the combined analysis, carriers of 2’–3’ high-risk factors had a nearly thirty-fold increased risk of early nephrotoxicity (OR 28.82; 3.43–241.83; *p* = 0.02) in comparison to the 0’–1’ group as reference (The second table in Section 2.7).

### 2.7. Association of SNPs and Clinical Factors with Hepatotoxicity

The risk of overall hepatotoxicity increased in the presence of *NR1/2* rs3732359 allele G (OR 2.06; 1.14–3.69; *p* = 0.016), postmenopausal age (OR 3.98; 1.77–8.92; *p* = 0.0007), and present metastases (OR 8.31; 2.87–8.32; *p* = 0.00008) (Table 7). In the cumulative analysis, we observed a gradation of hepatotoxicity risk from OR 8.64; 1.89–39.34; *p* = 0.005 in carriers of two high-risk factors to OR 66.0; 4.54–959.22; *p* = 0.01 in carriers of three high-risk factors. In the combined analysis, carriers of 1–3 unfavorable factors had an increased risk of overall hepatotoxicity (OR 5.57; 1.26–24.49; *p* = 0.022) compared to the reference group 0’. The comparison of the 3’ group to reference 0’s–2’s indicated an increased symptom risk to OR 14.22; 1.66–121.47; *p* = 0.015 (Table 8).

The *AKR1C3* rs32098968 genotype AA and present metastases were correlated with early hepatotoxicity (in the first two cycles of FAC chemotherapy) (OR 2.45; 1.08–5.54; *p* = 0.030 and OR 3.44; 1.19–9.94; *p* = 0.021) (Table 7). In the cumulative analysis, the carriers of two high-risk factors had an increased risk of early hepatotoxicity (OR 8.75; 1.90–40.17; *p* = 0.005). In the combined analysis, the concomitant presence of all two high-risk factors increased the risk of early hepatotoxicity (OR 5.88; 1.37–25.24; *p* = 0.016) (Table 8).

## 3. Discussion

This study was designed to establish a relationship between common SNPs in 3′UTR of ADME genes and clinical parameters as putative risk factors for myelosuppression, gastrointestinal side effects, nephrotoxicity, and hepatotoxicity during breast cancer chemotherapy. We explored ADME genes involved in pharmacokinetic of doxorubicin, 5′FU, and cyclophosphamide. Our results indicate that chemotherapy tolerance emerges from the simultaneous interaction of many genetic and clinical factors. In the following parts of the discussion, we discuss the results of this study divided into sections based on the functions of the analyzed genes and clinical parameters generating adverse effects of therapy.

### 3.1. Drug Metabolizers

In this study, we showed a correlation of genetic variants of genes *AKR1C3*, *ALDH5A1*, *UGT2B4, UGT2B15*, *DPYD*, *CYP1B1*, *NR1/2*, *SULT4A1*, and *NOS3* with bone narrow suppression, gastrointestinal side effects, hepatotoxicity, and nephrotoxicity. We report the significant correlation of *AKR1C3* rs3209896 AG with the risk of early nausea and bone narrow suppression, observed in our study as overall and early anemia. The AKR1C family proteins are enzymes involved in progesterone metabolism pathways [32]. Our results are consistent with previous studies that suggested the role of AKR1C3 in regulation of proliferation and differentiation of bone myeloid cells. Research indicates that reduced expression of AKR1C3 prevents the proliferation of human myeloid leukemia cells [33]. Genetic variants of *AKR1C3* gene were correlated with risk of lung and prostate cancer [34], leukemia [35], bladder cancer [36], B cell non-Hodgkin lymphoma, and breast cancer [37]. To the date, there are no data regarding the positive impact of the rs3209896 variant on cancer risk or treatment outcome. In the study of Liu et al. [35], it was not associated with the risk of childhood leukemia; similarly, the group of Voon reported a lack of correlation between *AKR1C3* rs3209896 genotype AG, survival time (PFS and OS), and chemotoxicity in breast cancer patients treated with doxorubicin-containing chemotherapy [38].

In our group of patients, the presence of *ALDH5A1* rs1054899 genotype AC increased the risk of early and recurrent severe neutropenia. Moreover, carriers of the AA genotype had an increased risk of recurrent nausea inducted by FAC chemotherapy. ALDH5A1 is involved in cyclophosphamide metabolic pathways [39], and its expression is a predictive factor for poor prognosis in early and advanced ovarian cancer [40]. The clinical importance of *ALDH5A1* rs1054899 has been confirmed in pharmacokinetic studies in Chinese patients with epilepsy treated with valproic acid (VPA) with anticonvulsant properties [41].

We report that SNPs in *UGT2* genes influenced the risk of myelotoxicity (rs1131878) and gastrointestinal symptoms (rs3100). The UGT2 belongs to phase II enzymes, playing important roles in toxicology and pharmacology, participating in clinically important drug–xenobiotic and drug–drug interactions. Dysfunction and/or changed expression of UGT is thought to contribute to interindividual differences in drug disposition, drug response, detoxification, and homeostasis, as well as to the risk of certain diseases [42]. Reported in the current study, *UGT2B4* rs1131878 common homozygote AA was the independent predictor of recurrent anemia while the presence of heterozygote AG was the independent predictor of recurrent neutropenia, including severe symptoms. In the literature, the presence of the *UGT2B4* rs1131878 heterozygote in comparison to both homozygotes increased the pancreatic cancer risk [43]. In pharmacogenomic and clinical studies, the *UGT2B4* rs1131878 heterozygote showed considerable, but not significant, associations with the metabolic ratio of risperidone in autism disorder patients [44]. Also, the pharmacogenetic study of Scherer’s group presented the correlations of *UGT2B4* rs1131878 heterozygotes with lower risk of colorectal cancer in users of nonsteroidal anti-inflammatory drugs than nonusers with only major alleles [45]. In our group of women, the *UGT2B15* rs3100 reference CC homozygote increased the risk of gastrointestinal side effects manifested by severe nausea. The rs3100 was reported to increase the risk of high-grade prostate cancer in African American men [46]. It is known that this variant is probably located within an miRNA binding site and, therefore, is crucial for the expression regulation of the *UGT2B15* gene. The exact mechanism of such influence is not yet elucidated, but the data in the literature suggest that several miRNAs (including miR-376c-3p), or perhaps indirectly another mechanism, regulate the influence of rs3100 on UGT2B15 expression [47].

In our study, the presence of the *DPYD* rs291593 minor T allele was an independent risk factor for leukopenia. Moreover, the CT heterozygote was a risk factor for recurrent nausea, while nephrotoxicity was linked to the presence of both homozygotes. The rare variant GG of the second studied *DPYD* SNP rs291592 was an unfavorable risk factor for early and recurrent leukopenia, and the allele G increased the risk of neutropenia and early nausea. DPYD is an enzyme of the 5-FU metabolism pathway [48]. In the literature, SNPs within DPYD are reported as the cause of enzyme deficiency, resulting in 5-FU therapy toxicity. Reduced DPYD activity contributes to interindividual differences in 5-FU therapies for efficacy and toxicity [49,50,51]. To date, the variants rs291592 and rs291593 have not been established as risk factors for chemotherapy toxicities. Conversely, the group of Etienne-Grimaldi did not confirm associations with digestive and/or neurologic and/or hematotoxicities and risk of severe toxicity under capecitabine in advanced breast cancer patients [52], although the clinical significance of rs291593 was shown by García-González et al., who reported worse disease-free survival of colorectal cancer patients carrying the GG homozygote when compared to other genotypes [53].

In our study, *CYP1B1* rs162562 common genotype AA modulated gastrointestinal side effects and early nausea risk. Cytochrome P450 1B1 (CYP1B1) is a monooxygenase involved phase I metabolism and is expressed in a variety of tumor tissues [54,55]. CYP1B1 plays the main role in the metabolism of estrogen and androgens substrates, and of a wide variety of xenobiotics. This enzyme catalyzes 4-hydroxyl-estrogens, which is a key reaction to hormonal carcinogenesis [56]. There are no reports describing the significant connection of rs162562 with clinical outcome of patients, while the data from disease-risk-assessing case–control studies are contradictory. The rs162562 polymorphism was studied by Gu et al., but they did not confirm associations with prostate cancer risk [57]. In the study of Burdon, however, rs162562 together with rs10916 were associated under a dominant model in the severe cases of primary open-angle glaucoma in a Caucasian population [58].

We observed the correlation of *GSTM3* rs3814309 rare CC genotype with an increased risk of early neutropenia. GSTM3 belongs to a family of phase II metabolizing enzymes that catalyze the conjugation of glutathione (GSH) to a wide variety of xenobiotic [56]. It plays an important role as antioxidant and has an important function in detoxification and clearance of reactive oxygen species in tumor tissues [59]. *GSTM3* polymorphisms are seen as risk factors for age-related and oxidative-stress-related diseases [60]. Also, polymorphic variants of the GSTs genes family are associated with an increased risk of cancer, including breast carcinoma [61]. The functional SNP *GSTM3* rs3814309 in the 3′UTR region is putatively associated with a few miRNAs interactions [24,62]. Moreover, the role of *GSTM3* rs3814309 was described as the switch of GSTM1 and GSTM4, which works by negative control of methylated sites. Unfortunately, despite the known function of *GSTM3* rs3814309, there are no data describing its exact clinical impact, whether on treatment efficiency or on cancer risk [63].

In our study, *NR1/2* rs3732359 allele G was associated with hepatotoxicity and recurrent vomiting. The transcription factor NR1/2 is an extremely important sensor of endo- and exogenous chemicals, which activates a variety of detoxification enzymes and pathways [64]. It should be noted that NR1/2 is the main mediator of expression of CYP3A4, the major metabolizer of drugs [65]. In the literature, *NR1/2* rs3732359 and rs3732360 were associated with higher CYP3A4 activity in vivo [66]. Polymorphisms in 3′UTR of *NR1/2* were also shown to increase resistance to chemotherapy in breast cancer patients, and the role of *NR1/2* rs3732359 was described in detoxification mechanisms of xenobiotics and in influence on bone marrow hematopoietic capacity [67,68,69,70]. In the study of Zeng, *NR1/2* rs3732359, together with several other *NR1/2* SNPs, significantly affected the voriconazole concentrations in patients with hematological malignancies and hematopoietic stem cell transplantation [71]. The group of Ren classified *NR1/2* rs3732359 together with *SLC15A1* rs2297322 and *FMO3* rs2266782 as multiple novel predictive biomarkers of docetaxel-induced myelosuppression specific to Chinese Han patients [72]. Another study underlined the importance of *NR1/2* rs3732359 to platelets and/or absolute neutrophil count (ANC) from baseline in cycle 1, and also to significant reduction in nadir hemoglobin, either dependent or independent of the effects on the pharmacokinetics of docetaxel in nasopharyngeal cancer patients [73]. Also, variants *NR1/2* rs3732359 and rs3732360 exhibited higher median oral midazolam clearance compared with homozygous reference genotypes for these SNPs [66,73].

In patients with *SULT4A1* rs138057 allele G, we observed higher risk of recurrent nausea, and severe and early severe vomiting. SULT4A1 is proven to be the doxorubicin-metabolizing enzyme [74]. This enzyme also works as potent metabolizer of endogenous chemicals, neurotransmitters, drugs, and xenobiotics. Its expression is known in cerebellum, hypothalamus, and cortex [75,76]. To our knowledge, the present study is the first to describe the associations of *SULT4A1* 3′UTR polymorphisms with breast cancer treatment outcome. Another gene variant identified in our study that increased the risk of early and recurrent nausea was *NOS3* rs2566508 genotypes TT/GG. In breast cancer tumors, it was positively associated with ER and PR status; also, SNPs in NOS3 in breast cancer patients have shown associations with invasive cancer and poor prognosis [77].

### 3.2. DNA Repair Genes

In the current study, *ERCC1* rs3212986 genotype GT and rs1046282 genotypes TC and CC determined a higher risk of hematologic toxicities. Moreover, *ERCC4* rs2276464 allele C increased the risk of recurrent nausea, while *ERCC4* rs4781563 genotype AA modulated the risk of early nephrotoxicity. The *ERCC1* rs3212986 genotype GT is crucial in drug-induced damage repair. In the literature, *ERCC1* rs3212986 is also described in association with treatment response, cancer risk, and overall survival [78,79,80,81] and with poor response to chemotherapy and shorter survival time of advanced NSCLC [82]. Moreover, this SNP is associated with the development of breast cancer in the Chinese population [83], and influences the response to chemotherapy and clinical outcome of gastric cancer [84]. Soares’ group noted that *ERCC1* rs3212986 is the risk factor for late gastrointestinal toxicity in cervical cancer patients. Recessive rs3212986 genotype AA presented a four-fold increased risk of developing late gastrointestinal toxicity compared to C allele carriers [5]. Goekkurts’ team noted that grade 3–4 anemia, leukopenia, and neutropenia were associated with polymorphisms within the *ERCC1* gene in gastroesophageal adenocarcinoma [85]. The pathogenicity of this variant may be due to changes in mRNA stability and dysfunction of the DNA repair process [5,86]. *ERCC1* polymorphism rs1046282 CC, identified in our study as a risk factor for early neutropenia, has been shown in the literature to have a more than two-fold lower risk of hepatocellular carcinoma [87]. In the same study, the occurrence of the rs1046282 TC genotype reduced the DNA load of hepatitis B virus and potentially affected the replication of HBV DNA in liver cancer patients.

Our results showed that two polymorphisms in the *ERCC4* gene were responsible for elevated risk of recurrent nausea (rs2276464) and early nephrotoxicity (rs4781563). To date, these results are not backed up by data in the literature; in fact, the lack of correlation of the rs2276464 variant with risk and prognosis of gastric, melanoma, and prostate cancer has been reported by few authors [59,88,89].

Our study indicates that hematologic, gastrointestinal side effects and kidney-related toxicity symptoms could be inducted by SNPs-related decrease in *ERCC1* and *ERCC4* expression. Polymorphisms located in the 3′UTR of *ERCC1* and *ERCC4* could modulate the major regulatory region of the gene and influence posttranscriptional modification of mRNA, which in turn affects the stability and function of mRNA, the ability of DNA repair is reduced, and cancer cells become prone to proliferation [87].

### 3.3. Drug Transporters

In this study, we evaluated several polymorphisms in selected membrane transporter genes as possible risk factors for treatment-induced toxicity. Strong influence was shown for variants in xenobiotic-transporting ATPase RALBP1, and in four genes of the ATP-binding cassette (ABC) superfamily (*ABCA1*, *ABCB1*, *ABCC1*, *ABCC5*) known for their impact on drug resistance [90].

We found a higher risk of recurrent leukopenia in carriers of *ABCA1* rs4149339 homozygotes CC and TT. Although the impact of this polymorphism on sensitivity to cytotoxic drugs has not been reported yet, the results of Zhao’s group indicate the functionality of said SNP. They reported an elevated risk of coronary artery disease with dyslipidemia in the Chinese population for the carriers of the *ABCA1* rs4149339 genotype CC [91]. In our study, two common polymorphisms in the *ABCC1* gene influenced FAC toxicity. The common homozygote GG of the rs129081 variant increased the risk of hematological-related symptoms: early and recurrent anemia and overall leukopenia (together with heterozygote GC). A correlation between *ABCC1* rs129081 and drug pharmacokinetics has not yet been reported, as shown by Steeghs and others in a study of telatinib in patients with solid tumors [92].

We reported a higher risk of recurrent neutropenia in carriers of *ABCC1* rs212091 genotype AG. Moreover, *ABCC1* rs212091 GG conditioned recurrent severe vomiting. Cao and colleagues reported that *ABCC1* rs212091 alleles GG and AG vs. AA had a lower risk of developing myelosuppression in Chinese acute myeloid leukemia patients. The functional significance of *ABCC1* rs212091 AG is correlated with creating a putative miRNA binding site for has-miR-1303 [93,94]. This polymorphic variant may result in adverse events during treatment or can potentially influence microsatellite instability in tumorigenesis, but functional studies are required [95]. Moreover, rs212091 is associated with virological failure in antiretroviral drugs therapy [96]. Polymorphism rs212091 AG was also associated with the protective function of the G allele and better prostate cancer survival. The data in the literature suggest that rs212091, together with rs35605, regulates RNA splicing of *ABCC1* and leads to the opposing effects on expression and function [97].

The results of our study showed a significant increased risk of recurrent severe and early neutropenia in carriers of the *ABCB1* rs17064 genotype AT. At the same time, the risk of recurrent and severe vomiting correlated with the presence of the minor T allele. Margier et al. described the association of *ABCB1* rs17064 with decreased fasting vitamin D status in a healthy male. This group speculated that rs17064 affects *ABCB1* expression, which may lead to altered vitamin D transport and status [98]. This SNP was examined in a few studies without associations with essential hypertension [99], depression, and antidepression response in Mexican Americans [100], ovarian outcome [101], and kidney allograft failure [102].

Our results showed that *ABCC5* rs3805114 reference allele AA was the only genetic factor independently associated with overall vomiting, accompanied by premenopausal age of patients who received FAC treatment. Also, the rs3805114 heterozygote AC was responsible for elevated risk of early nephrotoxicity. This SNP has been also associated with the ratio of CSF-to-plasma raltegravir concentration and exposure in healthy, HIV-negative adults of European descent [103]. However, Yuan et al. reported no relationship between this genotype and response in patients with chronic hepatitis B patients treated with entecavir [104].

The results of our study showed a higher risk of early anemia and early vomiting in carriers of *RALPB1* rs12680 allele C. The RALBP1 belongs to the xenobiotic-transporting ATPase family [105]. It is an antiapoptotic protein that mediates multidrug-resistance. RALBP1 is the major transporter of doxorubicin in lung cancer cells. In breast cancer cell lines, doxorubicin transport activity may be related to lower RALBP1 transporter activity [106]. To our knowledge, it is the first report of SNPs in *RALPB1* gene and chemotherapy outcome in breast cancer patients.

### 3.4. Clinical Determinants of FAC Chemotherapy Toxicity

Multiagent chemotherapy has a various range of side effects, including anemia [107]. These toxic myelosuppression manifestations increase risk of fatigue, malaise, and infection [108]. Bone narrow suppression is the dose-limiting factor because of potentially life-threatening neutropenia, leukopenia, and thrombocytopenia complications, i.e., severe infections and bleeding [109]. Moreover, hematological toxicity and side effects lead to limited practical application of drug and treatment delays. The FAC chemotherapy regimen induced grade I or II anemia in 55% of patients and grade II or IV in 11% of previously treated patients with metastatic breast cancer [110]. We observed anemia in 22.84% (8/35 cases) in breast cancer patients who underwent more than six cycles of the FAC regimen and 7.43% (20/269 cases) in the group of patients who underwent fewer than six cycles of the FAC regimen (Table 1). The first-mentioned group of patients had a nearly four times higher risk of overall anemia and a 15.43% increased risk of overall anemia in comparison to the second group. Concordant reports come from the study by Pourali et al., where anemia was observed in 38.2% of patients who underwent six cycles of chemotherapy and 50.9% in patients treated with eight cycles [107]. In addition, the group of Hassan et al. observed that the prevalence of anemia among solid tumor patients was increased after each chemotherapy cycle (after the first cycle anemia was observed in 0.5% patients, and four cycles of chemotherapy induced anemia in 43.6% patients) [111]. Furthermore, the danger of developing systemic toxicity during prolonged cancer chemotherapy is universally accepted, despite the improvement of outcomes and lowering of the recurrence rate for longer cycles. Such dependence has been shown by the group of Mayama, where the risk of hematological toxicities in endometrial cancer patients decreased with fewer treatment cycles [112].

Vomiting and nausea are some of the most common side effects reported during chemotherapy treatments. Chemotherapy-induced nausea and vomiting (CINV) is a common adverse effect that impacts treatment outcomes, quality of life, nutrition, ability to work, and treatment regimens [113,114]. Furthermore, risk factors reported in the literature for nausea and vomiting include chemotherapy type, female sex, history of morning or motion sickness, pregnancy-related nausea and vomiting, and history of low alcohol use [10,113,114,115,116,117]. In previous reports, the young age (<50 years) of cancer patients was one of the general risk factors for developing CINV [10,113,114,115,116]. In the present study, premenopausal age was a strong independent risk factor for nausea and vomiting. Similar observations came from the study of Nawa-Nishigaki et al., who observed that breast cancer patients younger than 55 years old treated with a anthracycline/cyclophosphamide regimen were at significant risk for both nausea and vomiting [118]. These data agree with the results of other studies conducted in group of premenopausal age women [119].

In our group of patients, women with advanced breast cancer were at higher risk of severe recurrent vomiting incidents. Tumor factor of TNM classification is crucial for treatment response [120]. In our report, the higher risk of gastrointestinal side effects correlated with tumor size over 20 mm in the greatest dimension (i.e., T component greater than T1), although the correlation of tumor size with recurrent severe vomiting is difficult to explain.

We report that the presence of metastases correlates with higher risk of hepatotoxicity. The presence of metastases is a key parameter of a poor prognosis and the main cause of mortality from breast cancer [121,122,123]. In our group, the liver was the fourth location of metastases, found in eight patients; therefore, the location itself cannot be considered responsible for the hepatic toxicity. One could assume that elevated liver enzyme levels in patients with advanced disease may reflect the activity of chemotherapy in multiple metastatic locations in the body.

We report that PR expression was an unfavorable factor correlated with recurrent neutropenia in breast cancer patients. Moreover, negative ER expression correlated with nephrotoxicity. The statuses of clinical factors like ER, PR, HER2, and tumor grade are the main prognostic determinants in breast cancer treatment. Progesterone receptor signaling pathways are strongly correlated with the estrogen receptor signaling pathway. The lack of progesterone receptor expression was reported as the factor for poor treatment outcome in breast cancer [124,125]. Lack of PR expression leads to estrogen receptor activation and the activation of pathways and cell proliferation [126,127]. ER, PR positivity, and HER2 gene negativity are associated with better prognosis in breast cancer patients. In turn, ER and PR negativity and HER2 positivity correlated with poor outcome and more advanced disease [128]. These factors determine prognosis of disease and the choice of optimal therapy. In our study, overexpression of HER2 was associated with neutropenia. Prognostic and predictive factor HER2 is overexpressed in 10–25% of breast cancers and is associated with a more aggressive form of breast cancer and poor outcome [129]. Neutropenia induced by chemotherapy is one of the basic problems in the treatment of cancer patients as it can lead to infections and increase the risk of discontinuing therapy.

In this study, the statistically significant cumulative models were constructed for seven symptoms of toxicities (Table 1, Table 2, Table 3, Table 4, Table 5, Table 6 and Table 7). The risk of adverse symptoms progressed with the growing number of genetic risk factors and/or clinical symptoms (Table 8). It should be noted that each model consisted of genetic factors belonging to at least two different functional groups, i.e., drug metabolizers, transporters, DNA repair machinery, and patients’ clinical characteristics. Thus, the elevated toxicity risk seen for the carriers of a maximum number of high-risk variants is the reflection of changes in the network of mechanisms crucial for FAC-drugs’ distribution and activity. In our results, it is also evident that the clinical factors’ impacts on toxicity risk do not prevail over the genetic ones, as could be expected. The clinical determinants are the components of over half of the constructed cumulative models (15 of 24) (Table 8), predominantly for gastrointestinal and kidney- and liver-related toxicities. The strongest clinical influences were seen in models of overall hepato- and nephrotoxicity, where, apart from the genetic factors, there were two clinical determinants that added to the symptoms’ risk. At the highest nephrotoxicity risk were the postmenopausal carriers of the variants in steroid hormones and 5FU metabolizers (*AKR1C3* and *DPYD*), with negative tumor ER expression. Even stronger dependence was detected for overall hepatotoxicity, where the only genetic factor (detox controller *NR1/2*) was accompanied by patients’ postmenopausal age and preexisting metastases. This could be, in part, a reflection of the group characteristics, as 32% of the metastases were located in the liver. The most complex model was constructed for recurrent nausea, where the presence of variants in four metabolizer genes (*SULT4A1*, *DPYD*, *NOS3*, *ALDH5A1*), together with the DNA repair (*ERCC4*), was responsible for high symptom risk. This result indicates the importance of those systems for the development of prolonged nausea. Surprisingly though, the cumulative model of recurrent nausea was the only one out of gastrointestinal side effects without the age-related compound. Similarly complex, but with regard to the presence of factors from all four functional groups, was the model of early nephrotoxicity risk.

The results of analyses focusing on the accumulation of high-risk factors on FAC toxicity symptoms confirmed the universally accepted multifactorial nature of those events. It is crucial to emphasize that in several models (early and recurrent anemia, recurrent nausea, early nephrotoxicity), there were any-symptom cases for the noncarriers of high-risk factors. At the same time, for models of early neutropenia and severe and severe recurrent vomiting, all patients carrying the maximum number of high-risk factors developed toxic symptoms. Those observations indicate that patients with the same diagnosis and treatment regime could be different in sensitivity to treatment. Patient selection based on genetic germline variants and clinical characteristics could be useful in tailoring treatment to improve its tolerance and effectiveness. Although the current FAC chemotherapy protocols are not flexible in terms of dose reduction, there are many premedication scenarios that could be implemented for patients at risk of treatment-related toxicity.

## 4. Materials and Methods

### 4.1. Patients and Samples

A total of 305 breast cancer subjects diagnosed from 1997 to 2012 were enrolled in Cancer Centre in Gliwice, Poland. The group was unified with respect to the treatment regimen. For all the women, the first-line chemotherapy regime was FAC, which combines doxorubicin (50 mg/m^2^), 5-fluorouracil (500 mg/m^2^), and cyclophosphamide (500 mg/m^2^). The drugs were administered intravenously on the first day of a 21-day cycle (six planned cycles). The chemotherapy was given in adjuvant or neoadjuvant setting. Peripheral blood (10 mL) from each patient was collected and stored (−80 °C) in EDTA-containing tubes. Genomic DNA used for genotyping studies was extracted using commercial DNA isolation kits. DNA samples were anonymized and stored at −20 °C. The characteristics of the breast cancer group are presented in Table 9.

Baseline blood test results obtained directly before the start of chemotherapy showed that the studied group was free of preexisting adverse symptoms. Women without germline mutations in *BRCA1* and *BRCA2* (HGVS: c.68_69del, c.181T>G, c.4034del, c.5266dup, c.5946del, c.9403del) genes were recruited for this study. All participants signed an informed consent form before being included in the study. The observations were carried out to 30 August 2017. This study was approved by the Bioethics Committee of the Institute of Oncology, Gliwice, Poland (no: KB/430-68/12), and was conducted in accordance with the Declaration of Helsinki.

The classification and grading of adverse treatment symptoms was based on EGOC Common Toxicity Criteria [130]. The frequency of FAC chemotherapy toxicities in our group is presented in Table 10.

Symptoms analyzed in this study were categorized by the following:Time of occurrence: overall—during whole first course of chemotherapy, and early—during the first two cycles.Severity: toxicity of any grade; severe—grades 3 and 4.Moreover, we analyzed symptoms during the first course of treatment: recurrent—during four or more cycles for any grade and recurrent severe events (grades 3 and/or 4) present at 2 or more cycles (Table 10).

Frequency of recurrent anemia was observed in 9 (2.96%), leukopenia 35 (11.48%), neutropenia 31 (10.20%), nausea 41 (13.49%), and vomiting 6 (1.98%) breast cancer women. Severe neutropenia was observed in 20 (6.56%), severe nausea in 9 (2.96%), and severe vomiting in 3 (0.99%) patients.

### 4.2. SNP Selection and Genotyping

The criteria for SNPs selection were as follows: location in 3′UTRs of ADME genes and the minor allele frequencies (MAF) ≥ 0.05 in the Caucasian population. Finally, we chose 33 SNPs in 23 genes involved in the metabolism and transport as well as the activity of cellular repair system and nuclear receptors in the FAC regimen based on anthracycline (doxorubicin), alkylate drug (cyclophosphamide), and antimetabolite (5′-fluorouracil). Selection of variants was based on an online database search: Ensembl BioMart release 105 December 2021 [131], PubMed [132], dbSNP [133]. SNPs analyses were designed using Primer3Plus [134] and Primer-BLAST [135], WatCut [136], or NEBCutter v2.0 [137] web-based tools. Alleles discrimination was detected by digestion PCR products and electrophoresis separation on agarose gel stained with ethidium bromide. All reactions were performed in accordance with the manufacturer’s instructions (EURx, Gdańsk, Poland or New England BioLabs, Ipswich, MA, USA). Sanger sequencing on selected samples (Genomed, Warsaw, Poland) was used to confirm genotype detection.. Details of the genetic variants, PCR primers sequences, amplification, and restriction fragments division conditions are available upon request or presented in [31].

### 4.3. Statistical Analyses and Study Design

Statistical analyses were preformed using Statistica v10.0 software (StatSoft Polska, Cracow, Poland). Hardy–Weinberg equilibrium (HWE) with the χ^2^ test was used to establish allele distributions and genomic variant frequencies.

Genetic factors were correlated with hematological toxicities (leukopenia, anemia, neutropenia), gastrointestinal side effects (vomiting, nausea), hepatotoxicity, and nephrotoxicity. Apart from genetic data, we were also looking for chemotherapy toxicity determinants among clinical and patient-related factors to obtain more comprehensive images. Thus, we included in our study the status of tumor receptors, TNM staging, tumor histotype, TNBC, patient’s age at the time of diagnosis, neo/adjuvant setting of chemotherapy, and number of cycles. Hormonotherapy, radiotherapy, immunotherapy, and follow-up events (death, metachronic breast cancer, recurrence, and progression) were excluded as presented after chemotherapy completion.

Statistical analyses for this study were designed in three steps:(1)Univariate analyses using Fisher two-way and Pearson exact tests were used to indicate a possible interdependence between genetic polymorphisms, clinical factors, and chemotherapy toxicities. A *p*-value ≤  0.10 was interpreted as a trend, and these results were moved to step 2.(2)Multivariate analyses were performed for each toxicity symptom with more than one factor from univariate analysis. In this step, logistic regression was used, with calculated odds ratios (ORs), 95% confidence intervals (95% CIs), and *p*-values. After stepwise regression, the sets of independent risk factors were possible to establish, or the multivariate model was rejected in the case of losing statistical significance (*p*  ≤   0.05). The sets of independent risk factors for a given toxicity symptom were then analyzed in step 3.(3)Cumulative analyses were performed to assess the simultaneous influence of many genetic and clinical factors on the appearance of treatment toxicity symptoms. Calculations were performed using logistic regression model odds ratios (ORs), 95% confidence intervals (95% CIs), and *p*-values. Patients with given toxicity symptoms were then divided into subgroups according to the number of independent risk factors they carried (groups 1’, 2’, etc.) The 0’s or noncarriers were defined as patients lacking high-risk factors, and were used as the control group in most analyses. The exception was made when toxicity symptoms were absent in the noncarriers group, and the control group was constructed from the 0’s and 1’s.

## 5. Conclusions

Our study highlights the importance of many factors from different regulatory levels and pathways of relationships leading to the manifestation of tolerance of breast cancer treatment. The SNPs selected for our analyses were located in the noncoding region of ADME genes and, with regard to main genetic principles, did not alter the protein synthesis. However, genetic variants located on the miRNA target side influence the expression of gene and regulation of mRNA degradation [138]. It should be emphasized that in our study we paid attention to the multifactorial determination of the occurrence of side effects or the toxicity of therapy. We suggest that the phenotypic occurrence of a symptom is determined by the accumulation of clinical and genetic factors and the patient’s condition. Our report presents an outline of the potential correlations between SNPs in 3′UTR ADME genes, clinical factors, and modulation of breast cancer treatment-related toxicities. We showed, in this study, that chemotherapy tolerance emerges from the simultaneous interaction of many genetic and clinical factors. The good tolerance of treatment with favorable outcomes seems to be the result of a delicate balance between drugs’ intake, excretion, metabolism rate, and innate patient, and also tumor, characteristics. Also, in the often-seen correlation of the given SNP with several toxic symptoms, the pleiotropic nature of genes involved in drugs’ management is strongly emphasized.

The effect of therapy depends on the interaction of many factors: clinical stage, histological type and its accompanying biomarkers [139], and genetic and epigenetic factors. Common variants in the coding sequences of genes could affect protein function and modulate treatment outcomes in cancer patients [140,141]. However, SNPs in noncoding regions (introns, 3′UTR, 5′UTR) of genes and regulatory factors may also participate in the response to therapy. This study is part of a publication cycle describing correlations among coding, noncoding 3′UTR SNPs, and clinical patient symptoms during FAC chemotherapy [31,142,143]. Being aware of the multifactorial nature of the response to therapy, we gradually expanded the scope of analyses in the same group of patients to include the role of multiple genetic and regulatory factors in the observed response to FAC therapy.

At the same time, our work has several limitations. The size of the studied group of patients is relatively small, which brings the possibility of false correlations and strongly reduces the statistical significance of the study. For the number of SNPs, the exact functional data are lacking, or only clinical correlations are reported, not necessarily in the oncology field; therefore, the interpretation of our results, to some extent, is brought down to theoretical deduction. For these SNPs, we tried to point out the available data from the literature that suggest, even indirectly, their modulatory effect on treatment outcome and/or tolerance in a variety of diseases. We hope that our work induces further studies that will shed light on the interplay between genetic variants and chemotherapeutics’ action in the cell and the body, externally seen as patients’ reaction to treatment. To start that, currently, our patient groups are undergoing the whole-genome sequencing procedure, technology that was not available for the project described in this and our previous papers [31,142,143], and additional SNPs in regulatory sequences of ADME genes and regulatory factors. We hope that such comprehensive genetic data will enable us to widen the set of genetic markers contributing to FAC regime toxicity. Also, given the pleiotropic nature of many genes, the potential new markers might not belong directly to FAC-metabolizing pathways, which could widen our understanding of drugs’ actions in the cells and organism.

## Figures and Tables

**Table 1 ijms-25-12283-t001:** Multivariate analysis of the associations between genetic and clinical factors and risk of anemia.

Anemia	Variable	FAC-Induced Anemia		Anemia Risk OR (±95% CI)	*p*
		Absent n (%)	Present n (%)		
Overall	*AKR1C3* rs3209896				
	**AA**/GG	148 (54.21)	8 (28.57)	1 (ref.)	
	AG	125 (45.79)	20 (71.43)	**3.00** (1.23–7.26)	**0.015**
	*ERCC1* rs3212986				
	**GG**/TT	173 (63.84)	12 (44.44)	1 (ref.)	
	GT	98 (36.16)	15 (55.56)	**2.50** (1.09–5.75)	**0.030**
	Number of chemotherapy cycles				
	0 (<6)	249 (90.22)	20 (71.43)		
	**1 (>6)**	27 (9.78)	8 (28.57)	**3.88** (1.48–10.16)	**0.005**
Early	*ABCC1* rs129081				
	CC/CG	102 (36.69)	10 (58.82)	1 (ref.)	
	**GG**	176 (63.31)	7 (41.18)	**3.52** (1.18–10.51)	**0.023**
	*AKR1C3* rs32098968				
	**AA**/GG	150 (53)	4 (23.53)	1 (ref.)	
	AG	133 (47)	13 (76.47)	**5.85** (1.56–21.83)	**0.008**
	*RALBP1* rs12680				
	**GG**	246 (87.23)	10 (62.50)	1 (ref.)	
	CC/CG	36 (12.77)	6 (37.50)	**4.41** (1.42–13.74)	**0.009**
Recurrent	*UGT2B4* rs1131878				
	AG/GG	138 (50.74)	14 (50.00)	1 (ref.)	
	**AA**	134 (49.26)	14 (50.00)	**8.91** (0.01–0.94)	**0.043**
	*ABCC1* rs129081				
	CG/CC	171 (63.81)	14 (50.00)	1 (ref.)	
	**GG**	97 (36.19)	14 (50.00)	**8.02** (0.02–0.63)	**0.012**
	*ERCC1* rs1046282				
	**TT**/CC	166 (61.25)	12 (42.86)	1 (ref.)	
	TC	105 (38.75)	16 (57.14)	**5.24** (1.0–6.99)	**0.047**

OR—odds ratio; 95% CI—confidence interval; bolded numbers indicate results with *p* < 0.05; bolded bases indicate reference genotype or factor.

**Table 2 ijms-25-12283-t002:** Multivariate analysis of the associations between genetic factors and risk of leukopenia.

Leukopenia	Variable	FAC-Induced Leukopenia		Leukopenia Risk OR (±95% CI)	*p*
		Absent n (%)	Present n (%)		
Overall	*ABCC1* rs129081				
	CC	28 (20.44)	20 (12.5)	1 (ref.)	
	CG/**GG**	109 (79.56)	140 (87.5)	**1.89** (0.27–0.99)	**0.048**
	*DPYD* rs291593				
	**CC**	75 (53.57)	67 (40.85)	1 (ref.)	
	CT/TT	65 (46.43)	97 (59.15)	**1.73** (1.08–2.76)	**0.020**
Early	*DPYD* rs291583				
	**AA**/AG	164 (84.10)	78 (72.22)	1 (ref.)	
	GG	31 (15.90)	30 (27.78)	**2.25** (1.25–4.05)	**0.006**
	*AKR1C3* rs32098968				
	AG	102 (52.85)	44 (40.74)	1 (ref.)	
	**AA**/GG	91 (40.74)	64 (59.26)	**1.77** (1.08–2.89)	**0.021**
Recurrent	*ABCA1* rs4149339				
	CT	98 (37.84)	6 (18.18)	1 (ref.)	
	**CC**/TT	161 (62.16)	27 (81.82)	**2.66** (0.14–0.95)	**0.040**
	*DPYD* rs291583				
	**AA**/AG	222 (82.22)	21 (61.76)	1 (ref.)	
	GG	48 (17.78)	13 (38.24)	**2.80** (1.26–6.19)	**0.011**

OR—odds ratio; 95% CI—confidence interval; bolded numbers indicate results with *p* < 0.05; bolded bases indicate reference genotype or factor.

**Table 3 ijms-25-12283-t003:** Multivariate analysis of the associations between genetic and clinical factors and risk of neutropenia.

Neutropenia	Variable	FAC-Induced Neutropenia		Neutropenia Risk OR (±95% CI)	*p*
		Absent n (%)	Present n (%)		
Overall	*DPYD* rs291583				
	**AA**	53 (36.81)	38 (23.75)	1 (ref.)	
	AG/GG	91 (63.19)	122 (76.25)	**1.83** (1.03–3.21)	**0.036**
	*ABCB1* rs17064				
	**AA**	256 (93.43)	27 (96.43)	1 (ref.)	
	AT	18 (6.57)	1 (3.57)	**4.56** (1.25–16.63)	**0.021**
	HER2 status				
	**0**	51 (45.13)	47 (33.57)	1 (ref.)	
	1	62 (54.87)	93 (66.43)	**1.69** (0.99–2.87)	**0.049**
Early	*GSTM3* rs3814309				
	CT/**TT**	179 (94.71)	93 (86.92)	1 (ref.)	
	CC	10 (5.29)	14 (13.08)	**3.12** (1.29–7.52)	**0.010**
	*ABCB1* rs17064				
	**AA**	186 (96.37)	96 (88.89)	1 (ref.)	
	AT	7 (3.63)	12 (11.11)	**3.52 (1.29–9.56)**	**0.013**
	*ERCC1* rs1046282				
	CT/**TT**	188 (98.43)	98 (91.59)	1 (ref.)	
	CC	3 (1.57)	9 (8.41)	**5.88** (1.50–22.97)	**0.010**
	*ALDH5A1* rs1054899				
	AA	104 (54.45)	47 (43.93)	1 (ref.)	
	AC/**CC**	87 (45.55)	60 (56.07)	**1.66** (1.00–2.75)	**0.046**
Recurrent	*ABCC1* rs212091				
	**AA**/GG	204 (75.00)	17 (56.67)	1 (ref.)	
	AG	68 (25.00)	13 (43.33)	**3.14** (1.36–7.25)	**0.007**
	*UGT2B4* rs1131878				
	**AA**/GG	152 (56.30)	10 (33.33)	1 (ref.)	
	AG	118 (43.70)	20 (66.67)	**2.68** (1.13–6.34)	**0.024**
	PR status				
	**0**	117 (45.17)	7 (25)	1 (ref.)	
	1	142 (54.83)	21 (75.00)	**2.65** (1.04–6.68)	**0.039**
Recurrent Severe	*ABCB1* rs17064				
	**AA**	268 (94.70)	16 (80.00)	1 (ref.)	
	AT	15 (5.30)	4 (20.00)	**5.13** (1.38–19.02)	**0.014**
	*UGT2B4* rs1131878				
	**AA**/GG	157 (58.87)	6 (30.00)	1 (ref.)	
	AG	124 (44.13)	14 (70.00)	**3.78** (1.26–11.24)	**0.016**
	*ALDH5A1* rs1054899				
	AA/**CC**	168 (59.79)	5 (26.32)	1 (ref.)	
	AC	113 (20.21)	14 (73.68)	**3.94** (1.34–11.53)	**0.012**

OR—odds ratio; 95% CI—confidence interval; bolded numbers indicate results with *p* < 0.05; bolded bases indicate reference genotype or factor.

**Table 4 ijms-25-12283-t004:** Multivariate analysis of the associations between genetic and clinical factors and risk of nausea.

Nausea	Variable	FAC-Induced Nausea		Nausea Risk OR (±95% CI)	*p*
		Absent n (%)	Present n (%)		
Early	*NOS3* rs2566508				
	TG	103 (64.78)	107 (76.43)	1 (ref.)	
	**TT**/GG	56 (35.22)	33 (23.57)	**1.73** (1.0–2.91)	**0.038**
	*CYP1B1* rs162562				
	CC/AC	89 (56.33)	94 (67.14)	1 (ref.)	
	**AA**	69 (43.67)	46 (32.86)	**1.63** (1.0–2.66)	**0.048**
	*DPYD* rs291583				
	**AA**	95 (58.64)	65 (46.10)	1 (ref.)	
	AG/GG	67 (41.36)	76 (53.90)	**1.90** (1.12–3.21)	**0.015**
	AGE				
	**other**	155 (95.68)	128 (90.14)	1 (ref.)	
	premenopausal	7 (4.32)	14 (9.86)	**2.98** (1.12–3.21)	**0.040**
Early Severe	*AKR1C3* rs3209896				
	**AA**/GG	150 (53.19)	5 (26.32)	1 (ref.)	
	AG	132 (46.81)	14 (73.68)	**3.80** (1.26–11.39)	**0.016**
	AGE				
	**other**	269 (94.39)	14 (73.68)		
	premenopausal	16 (5.61)	5 (26.32)	**7.49** (2.23–25.08)	**0.001**
Recurrent	*ERCC4* rs2276464				
	GG	153 (58.85)	16 (40.00)	1 (ref.)	
	**CC**/CG	107 (41.15)	24 (60.00)	**2.63** (1.26–5.48)	**0.01**
	*SULT4A1* rs138057				
	**AA**/AG	244 (95.69)	36 (87.80)	1 (ref.)	
	GG	11 (4.61)	5 (12.20)	**3.76** (1.16–12.17)	**0.027**
	*DPYD* rs291593				
	**CC**/TT	146 (55.73)	14 (34.15)	1 (ref.)	
	CT	116 (44.27)	27 (65.85)	**2.65** (1.27–5.59)	**0.01**
	*NOS3* rs2566508				
	TG	83 (32.17)	6 (14.63)	1 (ref.)	
	**TT**/GG	175 (67.83)	35 (85.37)	**2.74** (1.07–7.06)	**0.036**
	*ALDH5A1* rs1054899				
	**CC**/CA	244 (94.57)	34 (82.93)	1 (ref.)	
	AA	14 (5.43)	7 (17.07)	**4.72** (1.67–13.36)	**0.003**
Severe	*UGT2B15* rs3100				
	CT/TT	174 (65.66)	15(44.12)	1 (ref.)	
	**CC**	91 (34.34)	19 (55.88)	**2.31** (1.11–4.81)	**0.025**
	AGE				
	**other**	255 (94.44)	28 (82.35)	1 (ref.)	
	premenopausal	15 (5.56)	6 (17.65)	**3.52** (1.23–10.12)	**0.019**

OR—odds ratio; 95% CI—confidence interval; bolded numbers indicate results with *p* < 0.05; bolded bases indicate reference genotype or factor.

**Table 5 ijms-25-12283-t005:** Multivariate analysis of the associations between genetic and clinical factors and risk of vomiting.

Vomiting	Variable	FAC-Induced Vomiting		Vomiting Risk OR (±95% CI)	*p*
		Absent n (%)	Present n (%)		
Overall	*ABCC5* rs3805114				
	AC/CC	39 (17.81)	6 (7.79)	1 (ref.)	
	**AA**	180 (82.19)	71 (92.21)	**2.64** (1.05–6.62)	**0.037**
	AGE				
	**other**	215 (95.98)	68 (85.00)	1 (ref.)	
	premenopausal	9 (4.02)	12 (15.00)	**3.60** (1.38–9.42)	**0.009**
Early	*RALBP1* rs12680				
	**GG**	214 (88.07)	44 (77.19)	1 (ref.)	
	CC/CG	29 (11.93)	13 (22.81)	2.05 (0.97–4.32)	0.057
	AGE				
	**other**	234 (95.12)	50 (84.75)		
	premenopausal	12 (4.88)	9 (15.25)	**2.92** (1.11–7.65)	**0.028**
Recurrent	*ABCB1* rs17064				
	**AA**	278 (94.24)	4 (66.67)	1 (ref.)	
	AT/TT	17 (5.76)	2 (33.33)	**8.63** (1.37–54.56)	**0.021**
	*NR1/2* rs3732359				
	**AA**/AG	251 (85.96)	3 (50.00)	1 (ref.)	
	GG	41 (14.04)	3 (50.00)	**6.44** (1.19–34.77)	**0.030**
Severe	*ABCB1* rs17064				
	**AA**	271 (94.76)	12 (75.00)	1 (ref.)	
	AT/TT	15 (5.24)	4 (25.00)	**6.46** (1.65–25.25)	**0.007**
	*SULT4A1* rs138057				
	AG/GG	124 (44.29)	2 (12.50)	1 (ref.)	
	**AA**	156 (55.71)	14 (87.50)	**6.50** (1.37–30.76)	**0.017**
	AGE				
	**other**	271 (94.10)	12 (75.00)	1 (ref.)	
	premenopausal	17 (5.90)	4 (25.00)	**8.36** (2.07–33.75)	**0.003**
Early Severe	*SULT4A1* rs138057				
	**AA**	160 (55.94)	10 (90.91)	1 (ref.)	
	AG/GG	126 (44.06)	1 (9.09)	**11.34** (1.31–98.06)	**0.026**
	AGE				
	**other**	277 (94.22)	7 (63.64)	1 (ref.)	
	premenopausal	17 (5.78)	4 (36.36)	**14.50** (3.38–62.19)	**0.0003**
Severe Recurrent	*ABCC1* rs212091				
	**AA**/AG	293 (97.99)	2 (66.67)	1 (ref.)	
	GG	6 (2.01)	1 (33.33)	**36.17** (1.87–701.24)	**0.017**
	T (tumor; TNM component)				
	>1	222 (84.73)	1 (33.33)	1 (ref.)	
	**1**	40 (15.27)	2 (66.67)	**13.82** (0.98–195.27)	0.051

OR—odds ratio; 95% CI—confidence interval; bolded numbers indicate results with *p* < 0.05; bolded bases indicate reference genotype or factor.

**Table 6 ijms-25-12283-t006:** Multivariate analysis of the associations between genetic and clinical factors and risk of nephrotoxicity.

Nephrotoxicity	Variable	FAC-Induced Nephrotoxicity		Nephrotoxicity Risk OR (±95% CI)	*p*
		Absent n (%)	Present n (%)		
Overall	*DPYD* rs291593				
	CT	135 (49.27)	2 (14.29)	1 (ref.)	
	**CC**/TT	139 (50.73)	12 (85.71)	**7.23** (1.44–36.14)	**0.016**
	*AKR1C3* rs3209896				
	AG	137 (50.37)	2 (14.29)	1 (ref.)	
	**AA**/GG	135 (49.63)	12 (87.71)	**6.71** (1.33–33.75)	**0.020**
	AGE				
	**other**	212 (77.09)	5 (35.71)	1 (ref.)	
	postmenopausal	63 (22.91)	9 (64.29)	**7.57** (2.12–26.98)	**0.002**
	ER status				
	**positive**	116 (63.36)	5 (35.71)	1 (ref.)	
	negative	96 (36.64)	9 (64.29)	**5.71** (1.14–18.89)	**0.012**
Early	*ABCC5* rs3805114				
	**AA**	224 (85.82)	4 (50)	1 (ref.)	
	AC	37 (14.18)	4 (50)	**6.07** (1.11–32.99)	**0.035**
	*ERCC4* rs4781563				
	AG/**GG**	256 (96.97)	6 (75)	1 (ref.)	
	AA	8 (3.03)	2 (25)	**24.66** (2.22–273.1)	**0.008**
	*DPYD* rs291593				
	CT/TT	120 (44.78)	7 (87.5)	1 (ref.)	
	**CC**	148 (55.22)	1 (12.50)	**14.92** (1.13–195.99)	**0.038**
	AGE				
	**other**	80 (29.74)	7 (87.50)	1 (ref.)	
	perimenopausal	189 (70.26)	1 (12.50)	**10.25** (1.11–94.14)	**0.038**

OR—odds ratio; 95% CI—confidence interval; bolded numbers indicate results with *p* < 0.05; bolded bases indicate reference genotype or factor.

**Table 7 ijms-25-12283-t007:** Multivariate analysis of the associations between genetic and clinical factors and risk of hepatotoxicity.

Hepatotoxicity	Variable	FAC-Induced Hepatotoxicity		Hepatotoxicity Risk OR (±95% CI)	*p*
		Absent n (%)	Present n (%)		
Overall	*NR1/2* rs3732359				
	**AA**	99 (50.51)	35 (38.89)	1 (ref.)	
	AG/GG	97 (49.49)	55 (61.11)	**2.06** (1.14–3.69)	**0.016**
	AGE				
	other	211 (76.45)	19 (67.86)	1 (ref.)	
	**postmenopausal**	65 (23.55)	9 (32.14)	**3.98** (1.77–8.92)	**0.0007**
	M (metastases; TNM component)				
	**no**	225 (92.98)	19 (79.17)	1 (ref.)	
	yes	17 (7.02)	5 (20.83)	**8.31** (2.87–8.32)	**0.00008**
Early	*AKR1C3* rs32098968				
	AG/GG	127 (69.40)	16 (51.61)	1 (ref.)	
	**AA**	56 (30.6)	15 (48.39)	**2.45** (1.08–5.54)	**0.030**
	M (metastases; TNM component)				
	**no**	147 (92.45)	24 (77.42)		
	yes	12 (7.55)	7 (22.58)	**3.44** (1.19–9.94)	**0.021**

OR—odds ratio; 95% CI—confidence interval; bolded numbers indicate results with *p* < 0.05; bolded bases indicate reference genotype or factor.

**Table 8 ijms-25-12283-t008:** The association between accumulation of independent risk factors and toxicity risk.

Toxicity Symptom	Independent Risk Factors	Factors Number	Symptom Absent n (%)	Symptom Present n (%)	Toxicity Risk OR (±95% CI)	*p*
Anemia	*AKR1C3* rs3209896 AG *ERCC1* rs3212986 GT Number of chemotherapy cycles >6	0	74 (27.31)	4 (14.81)	1 (ref.)	
		1	149 (54.98)	9 (33.33)	1.12 (0.33–3.77)	0.857
		2	45 (16.61)	9 (33.33)	**3.70** (1.06–12.87)	**0.038**
		3	3 (1.11)	5 (18.52)	**30.83** (5.22–181.97)	**0.0001**
		0–2	268 (98.89)	22 (81.48)	1 (ref.)	
		3	3 (1.11)	5 (18.52)	**20.30** (4.52–91.18)	**0.00008**
Anemia Early	*ABCC1* rs129081 **GG** *AKR1C3* rs32098968 AG *RALBP1* rs12680CC/CG	0	76 (27.44)	0 (0.00)	1 (ref.)	
		1	140 (50.54)	5 (31.25)		
		2	56 (20.220	9 (56.25)	**6.94** (2.22–21.64)	**0.001**
		3	5 (1.81)	2 (12.50)	**17.28** (2.65–112.58)	**0.003**
		0–1	216 (77.98)	5 (31.25)	1 (ref.)	
		2–3	61 (22.02)	11 (68.75)	**7.79** (2.59–23.38)	**0.0002**
		0–2	272 (98.19)	14 (87.50)	1 (ref.)	
		3	5 (91.81)	2 (12.50)	**7.77** (1.37–43.95)	**0.020**
Anemia Recurrent	*UGT2B4* rs1131878 **AA** *ABCC1* rs129081 **GG** *ERCC1* rs1046282 TC	0	52 (18.18)	0 (0.0)	1 (ref)	
		1	133 (46.50)	1 (11.11)		
		2	84 (29.37)	3 (33.3)	6.61 (0.67–65.11)	0.104
		3	17 (5.94)	5 (55.56)	**54.41** (5.93–499.36)	**0.0004**
		0–1	185 (64.69)	1 (11.11)	1 (ref.)	
		2–3	101 (35.31)	8 (88.89)	**14.63** (1.79–119.85)	**0.012**
		0–2	269 (94.06)	4 (44.44)	1 (ref.)	
		3	17 (5.94)	5 (55.56)	**19.78** (4.83–80.91)	**0.00003**
Leukopenia	*ABCC1* rs129081 CG/**GG** *DPYD* rs291593 CT/TT	0	14 (10.22)	6 (3.75)	1 (ref.)	
		1	75 (54.74)	75 (46.88)	2.33 (0.84–6.44)	0.100
		2	48 (35.04)	79 (49.38)	**3.84** (1.37–10.76)	**0.010**
		0	14 (10.22)	6 (3.75)	1 (ref.)	
		1–2	123 (89.78)	154 (96.25)	**2.92** (1.09–7.86)	**0.033**
		0–1	89 (64.96)	81 (50.62)	1 (ref.)	
		2	48 (35.04)	79 (49.38)	**1.81** (1.13–2.89)	**0.013**
Leukopenia Early	*DPYD* rs291583 GG *AKR1C3* rs32098968 **AA**/GG	0	84 (43.52)	28 (26.17)	1 (ref.)	
		1	97 (50.26)	64 (59.81)	**1.97** (1.16–3.37)	**0.012**
		2	12 (6.22)	15 (14.02)	**3.75** (1.55–9.03)	**0.003**
		0	84 (43.52)	28 (26.17)	1 (ref.)	
		1–2	109 (56.48)	79 (73.83)	**2.17** (1.29–3.65)	**0.003**
		0–1	181 (93.78)	92 (85.98)	1 (ref.)	
		2	12 (6.22)	15 (14.02)	**2.45** (1.10–5.49)	**0.027**
Leukopenia Recurrent	*ABCA1* rs4149339 **CC**/TT *DPYD* rs291583 GG	0	80 (30.89)	4 (12.50)	1 (ref.)	
		1	151 (58.30)	18 (56.25)	2.38 (0.78–7.32)	0.127
		2	28 (10.81)	10 (31.25)	**7.14** (2.05–24.91)	**0.002**
		0	80 (30.89)	4 (12.50)	1 (ref.)	
		1–2	179 (69.11)	28 (87.50)	**3.13** (1.06–9.26)	**0.039**
		0–1	231 (89.19)	22 (68.75)	1 (ref.)	
		2	28 (10.81)	10 (31.25)	**3.75** (1.61–8.75)	**0.002**
Neutropenia	*DPYD* rs291583 AG/GG *ABCB1* rs17064 AT HER2 status positive	0	14 (12.39)	9 (6.52)	1 (ref.)	
		1	61 (53.98)	55 (39.86)	1.40 (0.56–3.52)	0.468
		2	37 (32.74)	65 (47.10)	**2.73** (1.07–6.99)	**0.034**
		3	1 (0.88)	9 (6.52)	**14.0** (1.38–142.44)	**0.020**
		0	14 (12.39)	9 (6.52)	1 (ref.)	
		1–3	99 (87.61)	129 (93.48)	2.03 (0.84–4.90)	0.115
		0–2	112 (99.12)	129 (93.48)	1 (ref.)	
		3	1 (0.88)	9 (6.52)	7.81 (0.096–63.28)	0.053
Neutropenia Early	*GSTM3* rs3814309 CC *ABCB1* rs17064 AT *ERCC1* rs1046282 CC *ALDH5A1* rs1054899 AC/**CC**	0	94 (50.00)	28 (26.92)	1 (ref.)	
		1	83 (44.15)	59 (56.73)	**2.38** (1.39–4.10)	**0.002**
		2	11 (5.85)	16 (15.38)	**4.88** (2.01–11.81)	**0.0004**
		3	0 (0.00)	1 (0.96)	--	--
		0	94 (89.52)	28 (62.22)		
		2–3	11 (10.48)	17 (37.78)	**5.18** (2.16–12.44)	**0.0002**
		0	94 (50.00)	28 (26.92)	1 (ref.)	
		1–3	94 (50.00)	73.08)	**2.71** (1.61–4.57)	**0.0002**
		0–1	177 (94.15)	87 (83.65)	1 (ref.)	
		2–3	11 (5.85)	17(16.35)	**3.14** (1.40–7.02)	**0.005**
Neutropenia Recurrent	*ABCC1* rs212091 AG *UGT2B4* rs1131878 AG PR status positive	0	46 (17.97)	1 (3.70)	1 (ref.)	
		1	117 (45.70)	6 (22.22)	2.36 (0.27–20.45)	0.433
		2	80 (31.25)	15 (55.56)	**8.63** (1.08–68.65)	**0.040**
		3	13 (5.08)	5 (18.52)	**17.69** (1.81–172.51)	**0.012**
		0	46 (17.97)	1 (3.70)	1 (ref.)	
		1–3	210 (82.03)	26 (96.30)	5.70 (0.75–43.43)	0.092
		0–2	243 (94.92)	22 (81.48)	1 (ref.)	
		3	13 (5.08)	5 (18.52)	**4.28** (1.38–13.08)	**0.011**
Neutropenia Recurrent Severe	*ABCB1* rs17064 AT *UGT2B4* rs1131878 AG *ALDH5A1* rs1054899 AC	0	91 (32.50)	1 (5.26)	1 (ref.)	
		1	129 (46.07)	6 (31.58)	4.23 (0.50–36.17)	0.185
		2	58 (20.71)	10 (52.63)	**15.69** (1.93–127.85)	**0.010**
		3	2 (0.71)	2 (10.53)	**91.0** (5.45–1520.36)	**0.002**
		0	91 (32.50)	1 (5.26)	1 (ref.)	
		1–3	189 (67.50)	18 (94.74)	**8.67** (1.13–66.48)	**0.037**
		0–2	278 (99.29)	17 (89.47)	1 (ref.)	
		3	2 (0.71)	2 (10.53)	**16.35** (2.15–124.32)	**0.007**
Nausea Early	*NOS3* rs2566508 **TT**/GG *CYP1B1* rs162562 **AA** *DPYD* rs291583 AG/GG AGE premenopausal	0	7 (4.43)	2 (1.43)	1 (ref.)	
		1	48 (30.38)	16 (11.43)	1.16 (0.21–6.38)	0.856
		2	61 (38.61)	66 (47.14)	3.78 (0.74–19.21)	0.100
		3	40 (25.32)	51 (36.43)	4.46 (0.64–15.76)	0.071
		4	2 (1.270	5 (3.57)	8.75 (0.86–23.13)	0.061
		0	7 (4.43)	2 (1.43)	1 (ref.)	
		1–4	151 (95.57)	138 (98.57)	3.19 (0.72–105.06)	0.151
		0–2	116 (73.42)	84 (60.00)	1 (ref.)	
		3–4	42 (26.58)	56 (40.00)	**1.84** (1.12–3.00)	**0.014**
		0–3	156 (98.73)	135 (96.43)	1 (ref.)	
		4	2 (1.27)	5 (3.57)	2.89 (0.55–15.23)	0.209
Nausea Early Severe	*AKR1C3* rs3209896 AG AGE premenopausal	0	138 (48.94)	4 (21.05)	1 (ref.)	
		1	140 (49.65)	11 (57.89)	2.71 (0.83–8.76)	0.094
		2	4 (1.42)	4 (21.05)	**34.5** (6.18–192.60)	**<0.00001**
		0	138 (48.94)	4 (21.05)	1 (ref.)	
		1–2	144 (51.06)	15 (78.95)	**3.59** (1.15–11.15)	**0.026**
		0–1	278 (98.58)	15 (78.95)	1 (ref.)	
		2	4 (1.42)	4 (21.05)	**18.5** (4.19–81.91)	**0.0001**
Nausea Recurrent	*ERCC4* rs2276464 **CC**/CG *SULT4A1* rs138057 GG *DPYD* rs291593 CT *NOS3* rs2566508 **TT**/GG *ALDH5A1* rs1054899 AA	0	24 (9.56)	--	1 (ref.)	
		1	87 (34.66)	7 (17.50)		
		2	98 (39.04)	11 (27.50)	1.78 (0.66–4.80)	0.252
		3	41(16.33)	21 (52.50)	**8.12** (3.12–20.66)	**0.00001**
		4	1 (0.40)	1 (2.50)	**15.86** (0.87–289.61)	**0.060**
		5	--	--	--	--
		0–1	111 (44.22)	7 (17.50)	1 (ref.)	
		2–4	140 (55.78)	33 (82.50)	**3.74** (1.59–8.80)	**0.0024**
		0–3	250 (99.60)	39 (97.50)	1 (ref.)	
		4	1 (0.40)	1 (2.50)	6.41 (0.39–105.84)	0.192
Nausea Severe	UGT2B15 rs3100 **CC** AGE premenopausal	0	166 (62.64)	13 (38.24)	1 (ref.)	
		1	93 (35.09)	17 (50.00)	**2.33** (1.08–5.03)	**0.030**
		2	6 (2.26)	4 (11.76)	**8.51** (2.11–34.33)	**0.009**
		0	166 (62.64)	13 (38.24)	1 (ref.)	
		1–2	99 (37.36)	21 (61.76)	**2.71** (1.29–5.67)	**0.008**
		0–1	259 (97.74)	30 (88.24)	1 (ref.)	
		2	6 (2.26)	4 (11.76)	**5.76** (1.53–21.67)	**0.009**
Vomiting	ABCC5 rs3805114 **AA** AGE premenopausal	0	37 (16.89)	5 (6.49)	1 (ref.)	
		1	175 (79.91)	63 (81.82)	**2.66** (0.98–7.10)	**0.049**
		2	7 (3.20)	9 (11.69)	**9.51** (2.37–38.17)	**0.0012**
		0	37 (16.89)	5 (6.49)	1 (ref.)	
		1–2	182 (83.11)	72 (93.51)	**2.92** (1.10–7.78)	**0.030**
		0–1	212 (96.80)	68 (88.31)	1 (ref.)	
		2	7 (3.20)	9 (11.69)	**4.01** (1.43–11.22)	**0.008**
Vomiting Early	*RALBP1* rs12680 CC/CG AGE premenopausal	0	203 (83.54)	40 (70.18)	1 (ref.)	
		1	39 (16.05)	13 (22.81)	1.69 (0.82–3.46)	0.148
		2	1 (0.41)	4 (7.02)	**20.3** (2.18–188.49)	**0.008**
		0	203 (83.54)	40 (70.18)	1 (ref.)	
		1–2	40 (16.46)	17 (29.82)	**2.15** (1.11–2.15)	**0.023**
		0–1	242 (99.59)	53 (92.98)	1 (ref.)	
		2	1 (0.41)	7.02	**18.26** (1.98–168.25)	**0.010**
Vomiting Recurrent	ABCB1 rs17064 AT/TT NR1/2 rs3732359 GG	0	236 (80.82)	2 (33.33)	1 (ref.)	
		1	54 (18.49)	3 (50.00)	**6.56** (1.06–40.50)	**0.042**
		2	2 (0.69)	1 (16.67)	**59.00** (3.63–959.44)	**0.004**
		0	236 (80.82)	2 (33.33)	1 (ref.)	
		1–2	56 (19.18)	4 (66.67)	**8.43** (1.50–47.50)	**0.015**
		0–1	290 (99.31)	5 (83.33)	1 (ref.)	
		2	2 (0.69)	1 (16.67)	**29.0** (2.22–378.35)	**0.010**
Vomiting Severe	ABCB1 rs17064 AT/TT SULT4A1 rs138057 **AA** AGE premenopausal	0	106 (37.86)	2 (12.50)	1 (ref.)	
		1	161 (57.50)	7 (43.75)	2.30 (0.47–11.38)	0.306
		2	13 (4.64)	6 (37.50)	**24.46** (4.39–136.26)	**0.0002**
		3	--	1 (6.25)	--	--
		0	106 (37.86)	2 (12.50)	1 (ref.)	
		1–3	174 (62.14)	14 (87.50)	**4.26** (0.94–19.25)	**0.058**
		0–1	263 (95.29)	9 (56.25)	1 (ref.)	
		2–3	13 (4.71)	7 (43.75)	**15.97** (5.12–49.87)	**<0.00001**
Vomiting Early Severe	*SULT4A1* rs138057 AG/GG AGE premenopausal	0	115 (40.21)	1 (9.09)	1 (ref.)	
		1	166 (58.04)	6 (54.55)	4.15 (0.48–35.30)	0.189
		2	5 (1.75)	4 (36.36)	**92** (8.42–1004.90)	**0.0002**
		0	115 (40.21)	1 (9.09)	1 (ref.)	
		1–2	171 (59.79)	10 (90.91)	6.72 (0.84–53.70)	0.071
		0–1	281 (98.25)	7 (63.64)	1 (ref.)	
		2	5 (1.75)	4 (36.36)	**32.11** (7.02–146.81)	**<0.00001**
Vomiting Severe Recurrent	ABCC1 rs212091 GG T (tumor; TNM) 1	0	216 (82.76)	1 (33.33)	1 (ref.)	
		1	45 (17.24)	1 (33.33)	4.80 (0.29–79.19)	0.271
		2	--	1 (33.33)	--	--
		0	216 (82.76)	1 (33.33)	1 (ref.)	
		1–2	45 (17.24)	2 (66.67)	**9.60** (0.84–109.36)	0.067
Nephrotoxicity	*DPYD* rs291593 **CT**/TT *AKR1C3* rs3209896 **AA**/GG AGE postmenopausal ER status negative	0	26 (10.08)	0 (0.00)	1 (ref.)	
		1	99 (38.37)	0 (0.00)		
		2	88 (34.11)	2 (14.29)		
		3	41 (15.89)	10 (71.43)	**25.98** (5.44–123.82)	**0.00004**
		4	4 (1.55)	2 (14.29)	**53.25** (5.85–484.25)	**0.0004**
		0–2	213 (82.56)	2 (14.29)	1 (ref.)	
		3–4	45 (17.44)	12 (85.71)	**28.40** (6.10–132.21)	**0.00001**
		0–3	254 (98.54)	12 (85.71)	1 (ref.)	
		4	4 (1.55)	2 (14.29)	**10.58** (1.75–64.12)	**0.010**
Nephrotoxicity Early	*ABCC5* rs3805114 AC *ERCC4* rs4781563 AA *DPYD* rs291593 **CC** AGE perimenopausal	0	83 (31.80)	0 (0.00)	1 (ref.)	
		1	127 (48.66)	1 (12.50)		
		2	42 (16.09)	2 (25.00)	10 (0.87–114.12)	0.063
		3	9 (3.45)	5 (62.50)	**4.88** (2.30–10.38)	**0.00003**
		0–2	252 (96.55)	3 (37.5)	1 (ref.)	
		3	9 (3.45)	5 (62.50)	**46.66** (9.55–227.80)	**<0.00001**
		0–1	210 (80.46)	1 (12.50)	1 (ref.)	
		2–3	51 (19.54)	7 (87.50)	**28.82** (3.43–241.83)	**0.002**
Hepatotoxicity	*NR1/2* rs3732359 AG/GG AGE postmenopausal M (metastases; TNM), present	0	22 (12.64)	2 (2.53)	1 (ref.)	
		1	95 (54.60)	27 (34.18)	3.13 (0.68–14.32)	0.139
		2	56 (32.18)	44 (55.70)	**8.64** (1.89–39.34)	**0.005**
		3	1 (0.57)	6 (7.59)	**66.0** (4.54–959.22)	**0.001**
		0	22 (12.64)	2 (2.53)	1 (ref.)	
		1–3	152 (87.36)	77 (97.47)	**5.57** (1.26–24.49)	**0.022**
		0–2	173 (99.43)	73 (92.41)	1 (ref.)	
		3	1 (0.57)	6 (7.59)	**14.22** (1.66–121.47)	**0.015**
Hepatotoxicity Early	*AKR1C3* rs32098968 **AA** M (metastases; TNM), present	0	105 (66.88)	12 (40.00)	1 (ref.)	
		1	48 (30.57)	14 (46.67)	**2.55** (1.09–5.96)	**0.029**
		2	4 (2.55)	4 (13.33)	**8.75** (1.90–40.17)	**0.005**
		0	105 (66.88)	12 (40.00)	1 (ref.)	
		1–2	52 (33.12)	18 (60.00)	**3.02** (1.35–6.79)	**0.007**
		0–1	153 (97.45)	26 (86.67)	1 (ref.)	
		2	4 (2.55)	4 (13.33)	**5.88** (1.37–25.24)	**0.016**

OR—odds ratio; 95% CI—confidence interval; bolded numbers indicate results with *p* < 0.05; bolded bases indicate reference genotype or factor.

**Table 9 ijms-25-12283-t009:** Characteristics of breast cancer patients group.

	Characteristics	n (%)
**General**	Age at diagnosis (years)	
	≤39	26 (8.0)
	40–50	222 (68.5)
	≥61	76 (23.5)
	Mean age at diagnosis in years (min–max)	54.7 (22.4–79.0)
	Year of diagnosis	
	1997–2004	15 (4.6)
	2005–2009	289 (89.2)
	2010–2012	20 (6.2)
	Histopathology	
	Invasive ductal carcinoma	230 (71.1)
	Invasive lobular carcinoma	22 (6.8)
	Carcinoma mixed type	6 (1.8)
	Other	29 (8.9)
	Unspecified	37 (11.4)
	Tumor grade	
	G1	39 (12.0)
	G2	71 (21.9)
	G3	83 (25.8)
	Bloom I	5 (1.5)
	Bloom II	12 (3.7)
	Bloom III	12 (3.7)
	Unspecified	102 (31.4)
**Receptors**	Estrogen receptor status	
	Negative	115 (35.5)
	Positive	190 (58.6)
	Unspecified	19 (5.9)
	Progesterone receptor status	
	Negative	133 (41.0)
	Positive	172 (53.1)
	Unspecified	19 (5.9)
	HER2 status	
	Negative	103 (31.8)
	Positive	167 (61.5)
	Unspecified	54 (16.7)
	triple-negative breast cancer (TNBC)	37 (11.4)
**TNM staging**	Tumor (T)	
	0	2 (0.6)
	1	43 (13.3)
	2	97 (29.9)
	3	52 (16.0)
	4	82 (25.3)
	Unspecified	50 (15.5)
	Nodes (N)	
	0	85 (26.2)
	1	108 (33.3)
	2	67 (20.7)
	3	19 (5.8)
	4	1 (0.3)
	Unspecified	44 (13.6)
	Metastases (M)	
	0	258 (79.6)
	1	25 (7.7)
	Unspecified	42 (12.7)
	Metastases locations	
	Liver	8 (32.0)
	Lungs	3 (12.0)
	Bones and lungs	3 (12.0)
	Bones	2 (8.0)
	Other	9 (36.0)
**Therapy**	Surgery	
	Amputation	187 (57.7)
	Conserving surgery. Including:	87 (26.8)
	With radicalization	14 (4.3)
	Without radicalization	73 (22.5)
	None	50 (15.5)
	Hormonotherapy	
	Yes	204 (63.0)
	No	120 (37.0)
	Immunotherapy (Herceptine)	
	Yes	36 (11.1)
	No	288 (88.9)
	Chemotherapy FAC	
	Adjuvant	136 (42.0)
	Neoadjuvant	188 (58.0)
	Mean number of cycles (range)	6.1 (3–9)
	Radiotherapy	
	Yes	265 (81.8)
	No	59 (18.2)
	Brachytherapy	7 (2.2)
	Mean radiation dose in Gy (range)	50.2 (20–70)
	Mean radiation dose in brachytherapy (range)	14 (10–30)
**Follow-Up**	Deaths	
	Yes	98 (32.2)
	No	207 (67.8)
	Median OS in months (min–max)	87.0 (4.3–177.7)
	Progression	
	Yes	107 (35.1)
	No	198 (64.9)
	Median PFS in months (min–max)	78.0 (0.9–176.7)
	Progression- locations of metastases	
	Bones	29 (27.1)
	Multiorgan spread	33 (30.9)
	Lungs	9 (8.4)
	Liver	8 (7.5)
	Lymph nodes	9 (8.4)
	Tumor growth	10 (9.3)
	Central nervous system	5 (4.7)
	Skin	3 (2.8)
	Eye socket	1 (0.9)
	Recurrence	
	Yes	14 (4.6)
	No	291 (95.4)
	Median RFS in months (min–max)	82.8 (0.5–176.7)
	Metachronous primary breast cancer	
	Yes	11 (3.6)
	No	294 (96.4)
	Median survival to next breast cancer diagnosis in months (min–max)	82.6 (0.5–176.7)
**Baseline Blood Tests Results**	White blood cells (10^3^/µL)	6.90 ± 1.92
	Neutrophiles (10^3^/µL)	4.04 ± 1.58
	Thrombocytes (10^3^/µL)	273.5 ± 74.10
	Monocytes (10^3^/µL)	0.690 ± 2.30 H
	Reticulocytes (10^3^/µL)	64.94 ± 20.67
	RDW (%)	14.59 ± 10.03 H
	Hemoglobin (g/dL)	13.80 ± 1.28
	Creatinine (µmol/L)	71.98 ± 14.00
	Bilirubin. Total (µmol/L)	10.76 ± 12.16
	ALAT (U/L)	21.13 ± 13.52
	AspAT (U/L)	21.07 ± 12.42
	ALP (U/L)	76.7 ± 40.78

HER2—human epidermal growth factor receptor-2; OS—overall survival; PFS—progression-free survival; RFS—recurrence-free survival; RDW—red (cell) distribution width; ALAT—alanine transaminase; AspAT—aspartate transaminase; ALP—alkaline phosphatase; H—high.

**Table 10 ijms-25-12283-t010:** Frequency of FAC chemotherapy toxicities by the ECOG grade.

Category	Symptom	Toxicity Grade, ECOG Common Toxicity Criteria n (%)					Number of Valid Cases
		0	1	2	3	4	
Overall toxicity: During the whole first course of treatment	Anemia	274 (90.13)	18 (5.92)	12 (3.95)	--	--	304
	Neutropenia	143 (46.88)	19 (6.23)	85 (27.87)	51 (16.72)	7 (2.30)	305
	Leukopenia	140 (45.90)	118 (38.69)	44 (14.43)	3 (0.98)	--	305
	Hepatotoxicity	201 (68.84)	84 (28.77)	6 (2.05)	1 (0.34)	--	292
	Gastrointestinal side effects—nausea	119 (39.02)	73 (23.93)	79 (25.90)	34 (11.15)	--	305
	Gastrointestinal side effects—vomiting	225 (73.77)	24 (7.87)	40 (13.11)	16 (5.25)	--	305
	Nephrotoxicity	274 (95.14)	11 (3.82)	3 (1.04)	--	--	288
Early toxicity: During first two cycles of treatment	Anemia	287 (94.72)	8 (2.64)	8 (2.64)	--	--	303
	Neutropenia	193 (63.70)	8 (2.64)	67 (22.11)	30 (9.90)	5 (1.65)	303
	Leukopenia	196 (64.47)	83 (27.30)	24 (7.89)	1 (0.33)	--	304
	Hepatotoxicity	251 (88.69)	30 (10.60)	2 (0.71)	--	--	283
	Gastrointestinal side effects—nausea	162 (53.29)	66 (21.71)	57 (18.75)	-	--	304
	Gastrointestinal side effects—vomiting	245 (80.33)	26 (8.52)	23 (7.54)	11 (3.60)	--	305
	Nephrotoxicity	269 (97.11)	5 (1.81)	3 (1.08)	--	--	277

## Data Availability

All data generated during this study are included in the published article. DNA sequences are available in NCBI SRA database under accession number PRJNA906438 (https://www.ncbi.nlm.nih.gov/sra/PRJNA906438, accessed on 30 November 2022).

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
