# Peer review of "Are the Common Genetic 3′UTR Variants in ADME Genes Playing a Role in Tolerance of Breast Cancer Chemotherapy?"

_ijms, 2024, doi:10.3390/ijms252212283_

Round 1
Reviewer 1 Report
Comments and Suggestions for Authors
The role of common genetic 3'UTR variants in ADME (Absorption, Distribution, Metabolism, and Excretion) genes in influencing the tolerance and response to breast cancer chemotherapy has been a subject of study. These variants can indeed play a role in how patients metabolize and respond to chemotherapy drugs, affecting both drug efficacy and the occurrence of adverse reactions. The authors indicated that chemotherapy tolerance emerges from the simultaneous interaction of genetic and clinical factors. The study identifies potential correlations between specific SNPs in the 3'UTR of ADME genes and the manifestation of toxic symptoms during breast cancer therapy.
However, the study’s limitations, particularly the small sample size and the lack of functional data, suggest that further research is needed to validate these findings. The proposed future direction of whole-genome sequencing is a promising step towards a deeper understanding of the genetic basis of chemotherapy-related toxicities.
I have some comments:
1. Prioritizing 3'UTR variants is useful, but it may also be a drawback because other genetic areas, including coding SNPs, intronic variants, or regulatory elements outside the 3'UTR, may also be important for drug toxicity and metabolism. Please discuss this point.
2. There are several punctuation errors in the text.
3. Please check the style of all references.
Author Response
Response to Reviewer 1
- Prioritizing 3'UTR variants is useful, but it may also be a drawback because other genetic areas, including coding SNPs, intronic variants, or regulatory elements outside the 3'UTR, may also be important for drug toxicity and metabolism. Please discuss this point.
Response: Additional text about other genetic areas including coding SNPs, intronic variants or regulatory elements outside the 3’UTR was added in Conclusion.
- There are several punctuation errors in the text.
Response: Corrected.
- Please check the style of all references
Response: Corrected.

Reviewer 2 Report
Comments and Suggestions for Authors
Please see the attached.

Author Response
Response to Reviewer 2
- Line 33-34, why Fraught with high risk is highlighted?
Response: Underlined section is the result of paste and edition error, corrected
- Line 83-84, do you have the publication related the previous study? If yes, please cite it here.
Response: Citation added
- Line 94, what does FAC means? This abbreviation firstly showed up here and would suggest to include its full name.
Response: FAC means the chemotherapeutic regime, containing: 5’-fluorouracil, anthracycline (doxorubicin) and cyclophosphamide. The FAC abbreviation with its meaning is given at the end of Introduction and in Materials and Methods: Patients and samples.
- Line 95, please use Odds ratios (ORs) when it showed up for the first time in the paper. What is the toxicity risk OR? And what range is considered toxicity risk levels? And there is a P-value was reported, please specify what groups were compared to get the P-values. How to obtain these data? Are all patients were separated into different groups to compare? Please include more details in statistical analyses or in the results part. It is quiet confusing in how the data present without any specification group comparisons. Highly suggest to include information about how the analysis was performed to obtain those data.
Response: The statistical methods, p-value threshold, meaning of abbreviations (OR, 95% CI) as well as detailed methodologic approach to study are explained in chapter Statistical analyses and study design. The citation regarding ECOG Common Toxicity Criteria added, whole section explaining treatment-related toxicities moved to chapter Patients and samples
- Line 97, table 1 includes too much information. Is there any efficient way to cite out the important information and then display it is a smaller table or figure? In addition, for FAC toxicity, what criteria is used for no/yes FAC Toxicity, respectively? This table is highly recommended to rearrange and additional information is need to make it easy to follow. Same for table 2. In table 2, what unfavorable factors refer to? Please specify.
Response: Table 1 has been divided into separate tables for each toxicity symptom. Heading of each table is rewritten in more clear way. Table 2 (Table 8 after revision) was designed and written in horizontal page orientation. The text for peer review appears to be in vertical setting as a whole, which makes harder to follow this table, so table 8 was edited to vertical. In Table 8 the column described by "p" was deleted. This column contained p values from distribution analysis (Fisher two-way and Pearson exact tests). Although the data is valid, the table presents assessment of risk calculated with regression analysis, and its p-value is presented. This step made this Table more accessible. Term “unfavourable” has been changed to “high risk”.
- Line 98, what are high-risk factors include in this study?
Response: Term “high risk factors” in cumulative analyses refers to factors that increase given symptom risk (Table 8). The header of factors describing column in Table 8 is changed to „Independent risk factors”
- Line 102, please explain the 0’-2’s reference group? Why this one used as reference group not others?
Response: The construction of groups in cumulative analyses has been explained in more specific manner in chapter Statistical analyses and study design.
- Line 107, what non-carrier group refers to?
Response: The construction of groups in cumulative analyses has been explained in more specific manner in chapter Statistical analyses and study design.
- Line 343-344, “We report that SNPs in UGT2 genes influenced risk of myelotoxicity (rs1131878) and gastrointestinal symptoms (rs3100).”Line 362-363, “The exact mechanism of such influence is not yet elucidated”.
Line 366 and line 374, “in our study the presence of DPYD rs291593 minor T allele was an independent risk factor of leukopenia.” “To the date the variants rs291592 and rs291593 was not established as risk factors of chemotherapy toxicities.” And further conclusion or potential mechanism that showing presence of DPYD rs291593 minor T allele was associate with leukopenia?
Line 381-382,” In our study, CYP1B1 rs162562 common genotype AA modulated gastrointestinal toxicity and early nausea risk.” Line 386-388, “The are no reports describing the significant
connection of rs162562 with clinical outcome of patients, while the data from disease risk assessing case-control studies are contradictory.” Typo is found in this sentence.
Line 393-394, “We observed correlation of GSTM3 rs3814309 rare CC genotype with an increased risk of early neutropenia.” Line 403-404, “Unfortunately, despite of the known function of GSTM3 rs3814309, there are no data describing its exact clinical impact, whether on the treatment efficiency or on cancer risk.”
The above statement including some important genotypes that might be related to myelotoxicity, leukopenia, gastrointestinal toxicity from this study and in each of the paragraph includes the citation of some studies to support these findings. However, it seems no direct support the findings, in this situation, please provide the gain of this study and the importance/necessary of these findings to make the statement stronger.
and
- Line 406-426, there are lots of references were included in this paragraph. The finding from this study saying that NR1/2 rs3732359 allele G was associated with hepatotoxicity and recurrent vomiting, however, some of the references in this paragraph seems not related to the finding from this study. Such as “The group of Ren classified NR1/2 rs3732359 together with SLC15A1 rs2297322 and FMO3 rs2266782 as multiple novel predictive biomarkers of docetaxel‐induced myelosuppression specific to Chinese Han patients. Another study underlines the importance of NR1/2 rs3732359 to platelets and/or absolute neutrophil count (ANC) from baseline in cycle 1, also to significant reduction in nadir haemoglobin, either dependent or independent of the effects on the pharmacokinetics of docetaxel in nasopharyngeal cancer patients.” It seems these citations were not associated with hepatotoxicity and recurrent vomiting. What is the purpose to use these citation? In addition, any other studies that can highly support the finding that NR1/2 rs3732359 allele G was associated with hepatotoxicity and recurrent vomiting? Any experimental data to support this point of view? There are multiple paragraphs has the same issue, please reconsider the important references that are really supported your statements.
Response to lines 343-426: In chapter Conclusion we pointed out the common problem with the interpretation of genomic data in relation to very specific clinical problem, meaning the frequent lack of exact matching references. It should be noted, that the 3’UTR variants were very rarely the subject of pharmacogenomics comparisons, or even the functional analyses, in opposition to variants in coding sequences. It is, in part, the reason to undertake such work. This situation impose, to certain extent, the use of references describing different diseases, treatment, etc., that enable the indirect reasoning. Where the exact matching references were available, they were used. The gain from this approach seems to be possible channeling the attention of researchers to not so commonly studied genetic variants, as the evidences of real impact on the treatment or patient as a whole are accumulating. The typo was corrected.
- Line 425-426, “In patients with SULT4A1 rs138057 allele G we observed higher risk of recurrent nausea, severe and early severe vomiting and recurrent nausea.” should be moved to the next paragraph in line 427.
Response: The sentence was moved to the next paragraph.
- Line 528, what is the % of increased risk of overall anemia in this study?
Response: The information was supplemented with data from Table 1.
- Line 582-584, grammar issue, please rewrite this sentence to make it clear. In addition, what are the 24 toxicities?
Response: The sentence was rewritten and supplemented with information from Tables 1-7. We corrected the number of toxicity symptoms as described in the Introduction.
- Line 605, what GI stands for? Please include the full name in line 605 as well as in table 4 description.
Response: Gastrointestinal side effects was included instead abbreviation GI in main text and in the Table 10. Also phrase “gastrointestinal toxicities” was replaced into “gastrointestinal side effects” in main text.
- Line 614-616, “Those observations indicate, that the patients who share the same diagnosis and treatment regime, could be divided into groups, which differ in expected oversensitivity to treatment.” Just want to know how patients were divided into different groups in this study.
Response: Patients in our study were not divided into groups with better or worse prognosis. Our study implicates that a prognostic model based on genetic factors and clinical characteristic could be introduced to assess the patient’s response to treatment, potentially as pre-treatment prediction tool.
- Line 622, patient and samples and Line 681, table 3, it seems that you are using the same patient resource from your previous published work (Genetic 3′UTR variations and clinical factors significantly contribute to survival prediction and clinical response in breast cancer patients) to report the new study. The data analysis description was similar but outcome is different. If that is the case, please specify what additional work in data analysis was performed to conclude these outcome. And compare to the previous work, what is the important of the extended work? And all the conclusions are based on the data analysis, any additional experiment to verify those genetic 3’UTR variants are associated with the symptoms of anemia and leucopenia?
Response: We used the same patients resource but statistical analyses and tests were different. In our previous study, we described a correlation between SNPs in 3’UTR of ADME genes and survival in breast cancer patients. In the current study, we report a correlation of germline SNPs in 3’UTRs of ADME genes and the risk of toxicity and side effects of breast cancer chemotherapy. The polymorphisms in breast cancer patients were tested in relation to 7 symptoms belonging to myelotoxicity, gastrointestinal side effects, nephrotoxicity and hepatotoxicity. High grade of toxicity was rare in our study, as the reflection of standardized dosage of drugs and patients’ premedication. According to chemotherapeutic guidelines severe toxicity is the reason to consider the pause of the treatment, until patient is well enough to resume it. It should be highlighted, that high grade toxicity is rarely and not directly reflected in shortened survival. Rather, the long-term toxicity (e.g. cardiotoxicity, ototoxicity or infertility) and consequent decreasing quality of life still remain to be problem in oncology. But still, our work focused on toxicity seen directly during the first course of treatment.
We did not do any additional experiments to verify genetic 3’UTR variants with the symptoms of anemia and leukopenia. The research financing plan did not assume performing functional tests.
- Line 628, what type of samples is referring to? Blood? Or tissue? Please specify. In addition, please include the experimental part in details in this part, such as how many material was taken for genomic DNA extraction? How is the sample handling for all samples?
Response: The information about material and sample handling for all samples were described more clearly in Materials and Methods: Patients and samples.
- Line 639, for FAC regimen, what are the frequency that those drugs were administered? What the does amount for different drugs? For the total 305 patients, they are in different years of diagnosis, different tumor grade, et al, that mean they are treated in different cycle of drugs and treatment, how to compare the data in patient to patient? Please specify how these data are compared among patients.
Response: In Materials and Methods following parts: Patients and samples, Statistical analyses and Study design description was significantly edited. We described steps of statistical analyses.

Round 2
Reviewer 2 Report
Comments and Suggestions for Authors
Thank you for answering all my questions. I don't have further comments.